# Sparse Gradient Compression for Fine-Tuning Large Language Models

## Abstract

Fine-tuning large language models (LLMs) for downstream tasks has become increasingly crucial due to their widespread use and the growing availability of open-source models. However, the high memory costs associated with fine-tuning remain a significant challenge, especially as models increase in size. To address this, parameter efficient fine-tuning (PEFT) methods have been proposed to minimize the number of parameters required for fine-tuning LLMs. However, these approaches often tie the number of optimizer states to dimensions of model parameters, limiting flexibility and control during fine-tuning. In this paper, we propose sparse gradient compression (SGC), a training regime designed to address these limitations. Our approach leverages inherent sparsity in gradients to compress optimizer states by projecting them onto a low-dimensional subspace, with dimensionality independent of the original model's parameters. By enabling optimizer state updates in an arbitrary low-dimensional subspace, SGC offers a flexible tradeoff between memory efficiency and performance. We demonstrate through experiments that SGC can decrease memory usage in optimizer states more effectively than exising PEFT methods. Furthermore, by fine-tuning LLaMA models on various downstream tasks, we show that SGC can deliver superior performance while substantially lowering optimizer state memory requirements, particularly in both data-limited and memory-limited settings.

## 1 Introduction

Large language models (LLMs) are increasingly being used across various disciplines, achieving remarkable performance in a wide range of natural language processing tasks. With the release of more open-source models, demand is growing to adapt them to downstream tasks (Touvron et al., 2023; Dubey et al., 2024). This is typically achieved using full fine-tuning, where all the parameters of a model are updated. However, as LLMs scale to billions of parameters, fine-tuning all the parameters of a model becomes increasingly challenging, demanding substantial memory resources.

Full fine-tuning requires not only storing billions of model weights, but also maintaining the gradients and optimizer states needed during training, which can drastically increase memory consumption (Chowdhery et al., 2022; Bai et al., 2023). For example, the Adam optimizer requires storing both the first-and second-order moments of the gradients, doubling the memory needed compared to storing the model's trainable parameters (Kingma & Ba, 2017). These memory constraints limit the practical ability to fine-tune LLMs, particularly in resource-constrained environments such as edge devices or personal computing platforms.

To address this problem, parameter efficient fine-tuning (PEFT) techniques have been introduced, to train a model using a significantly smaller number of parameters (Ding et al., 2023; Han et al., 2024). However, many existing methods lack the ability to provide both *flexible* and *granular* control over the number of optimizer states used for fine-tuning. Flexibility refers to the capacity to accommodate a broad range in the number of optimizer states, while granular control refers to the precision with which the number of optimizer states can be adjusted in small increments. This limitation may hinder the realization of a broader range of memory-performance tradeoffs, thereby restricting the potential of PEFT methods to achieve further efficiency gains.

On one end, we have approaches like BitFit (Zaken et al., 2022), which fine-tune on only the bias terms, using a minimal number of parameters, but is neither flexible nor offers granular control. On

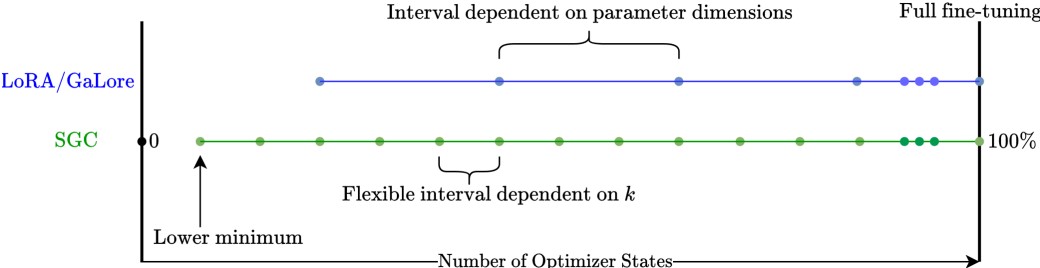

Figure 1: Diagram comparing SGC (green) and PEFT methods LoRA and GaLore (blue) in terms of the dimension of optimizer states compared to full fine-tuning. SGC enables a lower minimum and finer granularity for the number of optimizer states since it is independent of parameter dimensions.

the other hand, the popular low-rank adaptation (LoRA) is a more flexible approach that provides some control over the number of trainable parameters (Hu et al., 2021). However, there still exists limitations to both flexibility and granularity. LoRA reparameterizes the fine-tuned weight matrices $W^{(1)} \in \mathbb{R}^{m \times n}$ into $W^{(1)} = W^{(0)} + BA$, where $W^{(0)} \in \mathbb{R}^{m \times n}$ is the frozen pre-trained weight matrix, and $A \in \mathbb{R}^{r \times n}$ and $B \in \mathbb{R}^{m \times r}$ are two low-rank matrices of rank $r$ ($r \ll \min\{m, n\}$) to be trained. However, with LoRA, the number of optimizer states is a function of the dimensions of $A$ and $B$, which are dependent on $n$ and $m$, respectively. The minimum number of trainable parameters (achieved when $r = 1$) is equal to $n + m$, limited by the dimensions of $W^{(0)}$. Therefore, there exists a bound dependent on $n + m$ in which we cannot reduce the number of optimizer states during fine-tuning any further. Likewise, the granularity over parameters is also a function of $n$ and $m$, and notice that both flexibility and granularity are impacted negatively with larger models. Although a slightly different formulation might be needed, a similar limitation exists with many other approaches using prefix-tuning (Li & Liang, 2021) and gradient compression approaches, such as GaLore (Zhao et al., 2024) (see Appendix A).

To address the above limitation, we propose sparse gradient compression (SGC), a training regime that enables more flexible and granular control over the number of parameters to train during fine-tuning. SGC updates the optimizer states in a $k$-dimensional subspace, where $k$ is independent of the original parameters dimension and represents the number of optimizer states. This allows SGC to significantly reduce the number of optimizer states, irrespective of the pretrained model's size, with $k$ providing flexibility to balance performance and memory efficiency (see Figure 1). Importantly, this memory saving comes without sacrificing performance, as we will demonstrate in our experimental results.

The key idea behind SGC is leveraging the inherent sparsity of gradients during fine-tuning. By linearly projecting the optimizer states onto an arbitrarily lower-dimensional subspace, we can perform updates in this compressed space instead of the original space. A sparse recovery algorithm is then used to project the result of the optimizer function back into the original space, estimating the full-dimensional sparse vector from its lower dimensional representation, with sparsity originating from the gradients. By fine-tuning LLaMA2-7B, LLaMA3-8B, and LLaMa2-13B (Touvron et al., 2023; Dubey et al., 2024) on commonsense reasoning tasks, we show that SGC achieves comparable or better results than other PEFT methods while using a significantly smaller number of optimizer states. Additionally, we show that our approach yields improved fine-tuning performance in both data-limited and memory-limited scenarios.

## 2 RELATED WORKS

**Parameter Efficient Fine-tuning**. PEFT methods are used to reduce the expensive memory requirements for fine-tuning large models. Existing techniques can be split into several categories. Adapter-based methods introduce additional trainable modules that are inserted into the original frozen model (Houlsby et al., 2019; Pfeiffer et al., 2021; He et al., 2022; Mahabadi et al., 2021). However, these approaches can introduce latency during inference. Prompt tuning, on the other hand, adapts a model by adding learnable prefix tokens to the input (Li & Liang, 2021; Lester et al.,

2021; Liu et al., 2022). Despite their simplicity, these methods have structural limitations since they only train additional input tokens. LoRA is a widely used PEFT method that does not introduce additional inference latency (Hu et al., 2021). LoRA employs low-rank matrices to approximate the updates in the parameters during fine-tuning. Several variants of LoRA have been developed to either improve performance or further reduce the number of trainable parameters (Zhang et al., 2023; Xia et al., 2024; Liu et al., 2024; Kopiczko et al., 2024). Due to LoRA's popularity, extensive research has been conducted on both its theoretical foundations and empirical performance (Jang et al., 2024; Hayou et al., 2024; Mao et al., 2024). Additionally, quantization-based methods have been proposed to further reduce memory overhead Dettmers et al. (2023); Qin et al. (2024).

**Gradient Compression**. An area that has been relatively underexplored but is now gaining attention is gradient compression (Zhao et al., 2024; Hao et al., 2024; Liang et al., 2024; Wu et al., 2024; Song et al., 2024). These approaches selectively compress gradient information to reduce the size of optimizer states during training. One category of methods uses projection matrices to obtain a lower-rank gradients (Zhao et al., 2024; Hao et al., 2024; Liang et al., 2024). For instance, GaLore uses singular value decomposition (SVD) to obtain projection matrices (Zhao et al., 2024), while FLoRA utilizes random projection matrices (Hao et al., 2024). Liang et al. (2024) propose a method that updates the projection matrix in an online fashion using principal component analysis. Alongside projection matrices, gradient sparsity is another emerging factor. SIFT shows that gradients are approximately sparse, and achieves efficient fine-tuning by selecting parameters corresponding to the largest gradient magnitudes (Song et al., 2024). However, a significant limitation of this approach is that the selected parameters remain static, failing to fully capture the dynamic nature of gradient sparsity patterns during training.

## 3    PROBLEM FORMULATION

We investigate the task of updating the parameters of a neural network, $\boldsymbol{W} \in \mathbb{R}^d$, focusing specifically on *fine-tuning*, and without introducing any new weights into the model's architecture. The objective is to adapt pretrained weights $\boldsymbol{W}^{(0)} \in \mathbb{R}^d$ to $\boldsymbol{W}^{(1)} \in \mathbb{R}^d$ for a particular task.[1] The transition from $\boldsymbol{W}^{(0)}$ to $\boldsymbol{W}^{(1)}$ is defined as follows:

$$\boldsymbol{W}^{(1)} = \boldsymbol{W}^{(0)} + \Delta \boldsymbol{W}. \tag{1}$$

The parameter update process involves minimizing a loss function $\mathcal{L}$ with respect to $\boldsymbol{W}$ as follows:

$$\min_{\boldsymbol{W}} \mathcal{L}(\boldsymbol{W}^{(0)} + \Delta \boldsymbol{W}), \tag{2}$$

where we change the parameters in $\boldsymbol{W}$ minimizing $\mathcal{L}$ to achieve $\boldsymbol{W}^{(1)}$ from $\boldsymbol{W}^{(0)}$. With no closed form solution, the above problem is solved iteratively using the gradient signal $\boldsymbol{G}_t = \nabla_{\boldsymbol{W}_t} \mathcal{L} \in \mathbb{R}^d$ at every time step $t$, where $\boldsymbol{W}_t$ denotes the parameters in $\boldsymbol{W}$ at time $t$. Typically, to improve fine-tuning performance, an optimizer function $\rho_t(\cdot)$ is applied to the gradient $\boldsymbol{G}_t$, where $\rho_t$ requires storing and updating additional optimizer states, each with the same dimensions as $\boldsymbol{G}_t$. Therefore, the computational complexity and the memory requirements of applying the optimizer function is directly dependent on $d$, the dimension of $\boldsymbol{G}_t$.

With emergence of LLMs, $d$ has grown substantially large, making the execution of the optimizer function $\rho_t(\cdot)$ highly resource-intensive. To address this, we define a transformation function that reduces the dimension of $\boldsymbol{G}_t$ before being used in the optimizer function $\rho_t$. Specifically, we define $f : \mathbb{R}^d \to \mathbb{R}^k$ as the transformation function applied to the gradient $\boldsymbol{G}_t$ as $\hat{\boldsymbol{G}}_t = f(\boldsymbol{G}_t)$ for some $k \ll d$. Now we use $\hat{\boldsymbol{G}}_t$ as the input to the optimizer function $\rho_t$, reducing the dimension of the operations in the optimizer from a $d$-dimensional space to a $k$-dimensional space. The parameter update $\boldsymbol{W}$ for a single time step can be written as follows:

$$\boldsymbol{W}_{t+1} = \boldsymbol{W}_t - \eta g(\rho_t(\hat{\boldsymbol{G}}_t)), \tag{3}$$

where $\eta$ is the learning rate, and $g : \mathbb{R}^k \to \mathbb{R}^d$ is a transformation function that brings the output of $\rho_t$ back into the original $d$-dimensional space. We then denote the total changes in the parameters $\boldsymbol{W}$ after $T$ time steps as:

$$\boldsymbol{W}^{(1)} = \boldsymbol{W}^{(0)} - \eta \sum_t g(\rho_t(\hat{\boldsymbol{G}}_t)). \tag{4}$$

---

[1]Without loss of generality, we represent model parameters as vectors instead of matrices.

---

**Algorithm 1** AdamW at timestep $t$

---

1: **Inputs:** $\boldsymbol{G}_t, \beta_1, \beta_2, \epsilon$
2: $\boldsymbol{M}_t \leftarrow \beta_1 \boldsymbol{M}_{t-1} + (1 - \beta_1) \boldsymbol{G}_t$
3: $\boldsymbol{V}_t \leftarrow \beta_2 \boldsymbol{V}_{t-1} + (1 - \beta_2) \boldsymbol{G}_t^2$
4: $\boldsymbol{M}_t \leftarrow \frac{\boldsymbol{M}_t}{1 - \beta_1^t}$
5: $\boldsymbol{V}_t \leftarrow \frac{\boldsymbol{V}_t}{1 - \beta_2^t}$
6: $\boldsymbol{N}_t = \frac{\boldsymbol{M}_t}{\sqrt{\boldsymbol{V}_t} + \epsilon}$
7: **return** $\boldsymbol{N}_t$

---

This formulation allows us to perform the optimizer state updates in a smaller subspace $\mathbb{R}^k$ instead of the original space $\mathbb{R}^d$, where $k \ll d$. In practice, tracking the optimizer states in $\rho_t$ can be memory intensive if $k$ is large. Thus, the goal is to reduce $k$ as much as possible while maintaining a reasonable performance in minimizing $\mathcal{L}$.

## 4 METHODOLOGY

In this section, we introduce our proposed method for performing updates on a $k$-dimensional subspace. We begin by motivating our approach with an overview of the well known AdamW optimizer Kingma & Ba (2017); Loshchilov & Hutter (2019), followed by a detailed description of the gradient compression and decomposition processes. Additionally, we present two more efficient variants of the proposed approach, along with an analysis of the memory requirements.

### 4.1 MOTIVATION

Full fine-tuning model parameters $\boldsymbol{W}^{(0)}$ corresponds to the case where all parameters in $\boldsymbol{W}^{(0)}$ are updated, i.e., $f$ is the identity function and $\hat{\boldsymbol{G}}_t = \boldsymbol{G}_t$. If $\rho_t$ is also the identity function, i.e. we use no optimizer function, the updates simplify to stochastic gradient descent (SGD), and calculating $\Delta \boldsymbol{W}$ requires storing no optimizer states. However, using an optimizer function that makes use of momentum often yields better performance during fine-tuning. In this paper, we focus on the popular AdamW optimizer (see Algorithm 1), while both our formulation and proposed approach can be applied to various other optimizers. For full fine-tuning, AdamW requires storing two states $\boldsymbol{M}_t \in \mathbb{R}^d$ and $\boldsymbol{V}_t \in \mathbb{R}^d$ corresponding to the first and second moments, whose updates are controlled with hyperparameters $\beta_1 \in [0, 1]$ and $\beta_2 \in [0, 1]$, respectively. Taking this into consideration, the parameter update requires $2d$ memory in total for storing $\boldsymbol{M}_t$ and $\boldsymbol{V}_t$. We note that $(\cdot)^2$ and $\sqrt{\cdot}$ applied to vectors are elementwise square and square-root operations, and $\epsilon$ is a small constant to ensure numerical stability during division. With $g$ being the identify function, we have

$$\boldsymbol{W}_{t+1} = \boldsymbol{W}_t - \eta \boldsymbol{N}_t, \quad \boldsymbol{N}_t = \frac{\boldsymbol{M}_t}{\sqrt{\boldsymbol{V}_t} + \epsilon}. \tag{5}$$

Optimizer functions like AdamW contribute a large proportion of memory consumption during fine-tuning, and we will show how our approach aims to tackle this.

### 4.2 SPARSE GRADIENT COMPRESSION (SGC)

In full fine-tuning, the gradients being used as input in the AdamW algorithm can have a large dimension $d$. We would like to modify Algorithm 1 to update $\boldsymbol{M}_t$ and $\boldsymbol{V}_t$ on a $k$-dimensional subspace rather than the $d$-dimensional space, for some $k \ll d$, while retaining performance. This would significantly enhance the memory and compute efficiency of the optimizer, improving the efficiency of fine-tuning. We highlight that $\boldsymbol{M}_t$ and $\boldsymbol{V}_t$ are functions of $\boldsymbol{G}_t \in \mathbb{R}^d$ and $\boldsymbol{G}_t^2 \in \mathbb{R}^d$, respectively. Therefore, in order to perform the operations on $\boldsymbol{M}_t$ and $\boldsymbol{V}_t$ in a $k$-dimensional subspace, we need to represent $\boldsymbol{G}_t$ and $\boldsymbol{G}_t^2$ on that subspace. We make use of the observation that $\boldsymbol{G}_t$ is a quasi-sparse vector (Song et al., 2024) and can be compressed to a lower dimensional subspace to reduce memory usage in the optimizer function since both $\boldsymbol{M}_t$ and $\boldsymbol{V}_t$ can also be represented in the lower dimensional subspace. This enables us to conduct fine-tuning with much greater efficiency and control over the memory usage.

We first sparsify $\boldsymbol{G}_t \in \mathbb{R}^d$ by keeping only $s$ non-zero elements corresponding to $s$ entries with largest magnitudes, and set all other elements to zero which is denoted by $\text{Sparsify}_s(\cdot)$. The sparsified gradient is then projected onto a lower dimensional subspace of an arbitrary dimension $k$ using a projection matrix $\boldsymbol{A} \in \mathbb{R}^{k \times d}$ that is initialized before fine-tuning:

$$\tilde{\boldsymbol{G}}_t = \text{Sparsify}_s(\boldsymbol{G}_t) \in \mathbb{R}^d, \quad \boldsymbol{p}_t = \boldsymbol{A}\tilde{\boldsymbol{G}}_t \in \mathbb{R}^k. \tag{6}$$

To compress $\boldsymbol{G}_t^2$, we use the fact that elementwise squares retain the sparsity pattern of $\boldsymbol{G}_t$. Thus, similar to $\boldsymbol{G}_t$, we can represent $\boldsymbol{G}_t^2$ on the $k$-dimensional subspace through

$$\boldsymbol{q}_t = \boldsymbol{A}\tilde{\boldsymbol{G}}_t^2 \in \mathbb{R}^k. \tag{7}$$

With $\boldsymbol{G}_t$ and $\boldsymbol{G}_t^2$ represented in a compressed form with dimension $k$ as $\boldsymbol{p}_t$ and $\boldsymbol{q}_t$, respectively, we modify Algorithm 1 by representing $\boldsymbol{M}_t$ and $\boldsymbol{V}_t$ in this $k$-dimensional subspace as follows:

$$\boldsymbol{M}_t \leftarrow \beta_1 \boldsymbol{M}_{t-1} + (1 - \beta_1)\boldsymbol{p}_t, \tag{8}$$
$$\boldsymbol{V}_t \leftarrow \beta_1 \boldsymbol{V}_{t-1} + (1 - \beta_1)\boldsymbol{q}_t. \tag{9}$$

Accordingly, we can perform the updates on optimizer states $\boldsymbol{M}_t$ and $\boldsymbol{V}_t$ on a $k$-dimensional subspace since $\boldsymbol{p}_t$ and $\boldsymbol{q}_t$ are $k$-dimensional. However, we need to go back to the original $d$-dimensional space to perform the weight updates from $\boldsymbol{W}_t$ to $\boldsymbol{W}_{t+1}$. As indicated in 3, this transform is conducted using the function $g : \mathbb{R}^k \to \mathbb{R}^d$. Rewriting 4, this problem is equivalent to finding a function $g(\cdot)$ to perform the update

$$\boldsymbol{W}^{(1)} = \boldsymbol{W}^{(0)} - \eta \sum_t g(\rho_t(\boldsymbol{p}_t, \boldsymbol{q}_t)). \tag{10}$$

Thus, this approach enables performing the updates on a $k$-dimensional subspace instead of the $d$-dimensional space using AdamW. The only missing part is how to define $g(\cdot)$ that enables going from a $k$-dimensional subspace back to the original $d$-dimensional space for the parameter updates. Next, we introduce an approach to achieve such $g(\cdot)$ functionality.

### 4.3 Compressed Sensing of Optimizer States

Ideally, we would like to use $\boldsymbol{G}_t$ and $\boldsymbol{G}_t^2$ or their respective sparse versions $\tilde{\boldsymbol{G}}_t$ and $\tilde{\boldsymbol{G}}_t^2$ for the optimizer algorithms; however, for enhancing efficiency we instead use $\boldsymbol{p}_t$ and $\boldsymbol{q}_t$. We note that $\boldsymbol{p}_t$ and $\boldsymbol{q}_t$ are the results of linear projection of sparse vectors $\tilde{\boldsymbol{G}}_t$ and $\tilde{\boldsymbol{G}}_t^2$, respectively, onto a $k$-dimensional subspace. Thus, function $g(\cdot)$ should provide a good estimate of $\tilde{\boldsymbol{G}}_t$ and $\tilde{\boldsymbol{G}}_t^2$ when applied to $\boldsymbol{p}_t$ and $\boldsymbol{q}_t$, respectively. As a result, the problem is to estimate the sparse vectors $\tilde{\boldsymbol{G}}_t$ and $\tilde{\boldsymbol{G}}_t^2$ from their compressed form $\boldsymbol{p}_t$ and $\boldsymbol{q}_t$, respectively, compressed with linear projection.

We use a recovery algorithm from compressive sensing (CS) to achieve the function $g(\cdot)$, which aims to estimate a sparse vector from its compressed form, compressed through linear projection. CS is a signal processing technique used to recover signals using fewer measurements than the Nyquist rate, when the signals are sparse (Candes et al., 2004; Donoho, 2006). Consider an $s$-sparse signal $\boldsymbol{x} \in \mathbb{R}^d$ with $s$ non-zero entries. We can reconstruct $\boldsymbol{x}$ from a set of linear measurements $\boldsymbol{y} = \boldsymbol{A}\boldsymbol{x}$, if the measurement matrix $\boldsymbol{A} \in \mathbb{R}^{k \times d}$ satisfies the restricted isometry property (RIP) for some number of measurements $k \leq d$ (Candes & Tao, 2005; Candes, 2008). The RIP conditions can be satisfied with high probability if every element of $\boldsymbol{A}$ is independent and identically distributed according to a zero-mean normal distribution with standard deviation $1/\sqrt{k}$, and $k \geq \kappa s$, where $\kappa$ is an algorithm dependent constant (Candes et al., 2004).

There exist various recovery algorithms to recover the $d$-dimensional $s$-sparse signal $\boldsymbol{x}$ from measurements $\boldsymbol{y}$ (Marques et al., 2018). In this paper, we use a greedy algorithm named orthogonal matching pursuit (OMP) (Pati et al., 1993). To enhance efficiency, inspired by Zhu et al. (2020), we have developed a GPU optimized version of OMP, enabling its seamless integration with fine-tuning (see Appendix B for details). The OMP algorithm reconstructs an $s$-sparse vector $\boldsymbol{x}$ from the measurements $\boldsymbol{y}$ having knowledge about the measurement matrix $\boldsymbol{A}$ denoted as follows:

$$\hat{\boldsymbol{x}} = \text{OMP}_{\boldsymbol{A}}(\boldsymbol{y}). \tag{11}$$

We now apply the recovery algorithm OMP to map the updates $\boldsymbol{M}_t$ and $\boldsymbol{V}_t$, given in equations 8 and 9, respectively, from the $k$-dimensional subspace back to the original $d$-dimensional space. With the

---

**Algorithm 2** SGC at timestep $t$

---

1: **Inputs:** $\boldsymbol{G}_t, \boldsymbol{A}, s, \beta_1, \beta_2, \epsilon$
2: $\boldsymbol{p}_t = \boldsymbol{A} \operatorname{Sparsify}_s(\boldsymbol{G}_t), \quad \boldsymbol{q}_t = \boldsymbol{A} \operatorname{Sparsify}_s(\boldsymbol{G}_t^2)$
3: $\boldsymbol{M}_t \leftarrow \beta_1 \boldsymbol{M}_{t-1} + (1 - \beta_1) \boldsymbol{p}_t$
4: $\boldsymbol{V}_t \leftarrow \beta_2 \boldsymbol{V}_{t-1} + (1 - \beta_2) \boldsymbol{q}_t$
5: $\boldsymbol{M}_t \leftarrow \frac{\boldsymbol{M}_t}{1 - \beta_1^t}$
6: $\boldsymbol{V}_t \leftarrow \frac{\boldsymbol{V}_t}{1 - \beta_2^t}$
7: $\boldsymbol{N}_t = \alpha \frac{\operatorname{OMP}_{\boldsymbol{A}}(\boldsymbol{M}_t)}{\sqrt{\operatorname{OMP}_{\boldsymbol{A}}(\boldsymbol{V}_t) + \epsilon}}$
8: **return** $\boldsymbol{N}_t$

---

initialization $\boldsymbol{M}_0 = \boldsymbol{0}$ and $\boldsymbol{V}_0 = \boldsymbol{0}$, we can rewrite the updates $\boldsymbol{M}_t$ and $\boldsymbol{V}_t$ as:

$$\boldsymbol{M}_t = \boldsymbol{A} \sum_{i=1}^{t} h_i(\beta_1) \tilde{\boldsymbol{G}}_i, \quad \boldsymbol{V}_t = \boldsymbol{A} \sum_{i=1}^{t} h_i(\beta_2) \tilde{\boldsymbol{G}}_i^2 \tag{12}$$

where $h_i(\cdot)$ is a constant only a function of $\beta_1$ or $\beta_2$. We observe that $\sum_{i=1}^{t} h_i(\beta_1) \tilde{\boldsymbol{G}}_i$ and $\sum_{i=1}^{t} h_i(\beta_2) \tilde{\boldsymbol{G}}_i^2$ are linear combinations of the first and second moments of the sparsified gradients, respectively. Assuming that the total changes in the sparsity of $\boldsymbol{G}_t$ over all $t$ can be bounded by some constant $\tilde{s} \ll d$, we can use the OMP algorithm as in 11 to almost accurately recover the original $d$-dimensional representations of $\boldsymbol{M}_t$ and $\boldsymbol{V}_t$. After applying OMP to $\boldsymbol{M}_t$ and $\boldsymbol{V}_t$ separately, we obtain $\boldsymbol{N}_t$ as follows:

$$\boldsymbol{N}_t = \alpha \frac{\operatorname{OMP}_{\boldsymbol{A}}(\boldsymbol{M}_t)}{\sqrt{\operatorname{OMP}_{\boldsymbol{A}}(\boldsymbol{V}_t) + \epsilon}}, \tag{13}$$

where $\alpha$ is a scaling factor. We note that the feasibility of obtaining $\boldsymbol{N}_t$, as in 13, is ensured by the fact that $\tilde{\boldsymbol{G}}_t$ and $\tilde{\boldsymbol{G}}_t^2$, and thus $\boldsymbol{M}_t$ and $\boldsymbol{V}_t$, share the same sparsity pattern. Consequently, the indices of the non-zero entries in $\operatorname{OMP}_{\boldsymbol{A}}(\boldsymbol{M}_t)$ and $\operatorname{OMP}_{\boldsymbol{A}}(\boldsymbol{V}_t)$ are identical. Furthermore, the sparsity level $s$ provides a tradeoff between performance and efficiency. Clearly, a larger $s$ leads to better performance since $\tilde{\boldsymbol{G}}_t$ provides a better estimate for $\boldsymbol{G}_t$; however, it increases the computational overhead with the OMP algorithm in recovering an $s$-sparse vector.

Following compression, the optimizer states $\boldsymbol{M}_t$ and $\boldsymbol{V}_t$ are now $k$-dimensional vectors. Setting $k = \kappa s$ leads to a reasonable recovery of $\sum_{i=1}^{t} h_i(\beta_1) \tilde{\boldsymbol{G}}_i$ and $\sum_{i=1}^{t} h_i(\beta_2) \tilde{\boldsymbol{G}}_i^2$ from $\boldsymbol{M}_t$ and $\boldsymbol{V}_t$ in 12, using OMP. Now, the size of the optimizer states in AdamW becomes purely a function of $k$, and can be controlled at a granular level.

We refer to our proposed method as SGC, which uses the AdamW optimizer and is presented in Algorithm 2. For ease of presentation, we represent this algorithm with $\boldsymbol{N}_t = \operatorname{SGC}(\boldsymbol{G}_t)$, which takes the gradient vector $\boldsymbol{G}_t \in \mathbb{R}^d$ as the input and outputs $\boldsymbol{N}_t \in \mathbb{R}^d$, while the optimizer states $\boldsymbol{M}_t$ and $\boldsymbol{V}_t$ are $k$-dimensional. Incorporating this into our formulation in 4 yields:

$$\boldsymbol{W}^{(1)} = \boldsymbol{W}^{(0)} - \eta \sum_t \operatorname{SGC}(\boldsymbol{G}_t). \tag{14}$$

## 4.4 EFFICIENT SGC

Here, we propose two efficient alternatives of the SGC algorithm.

**Memory Efficient SGC (MESGC).** Based on our observations, size of the projection matrix $\boldsymbol{A} \in \mathbb{R}^{k \times d}$ may significantly contribute to the computation overhead. Although it is initialized only once before fine-tuning, the memory requirements can become substantial depending on the value of $s$, the sparsity level of $\tilde{\boldsymbol{G}}_t$, particularly when applying the OMP algorithm. To address this issue, we introduce the idea of chunking the gradient signals prior to applying a projection matrix. Specifically, we split $\boldsymbol{G}_t$ into $c$ equal sized chunks before sparsifying and projecting each chunk. This enables the projection matrix $\boldsymbol{A}$ to be much smaller in size from $k \times d$ to $(k \times d)/c$. We split $\boldsymbol{G}_t$ to $c$ equal-size chunks $\boldsymbol{G}_t = [\boldsymbol{G}_t^1, \ldots, \boldsymbol{G}_t^c]$ and apply the SGC algorithm to each $\boldsymbol{G}_t^i$. Accordingly, we have $\boldsymbol{N}_t^i = \operatorname{SGC}(\boldsymbol{G}_t^i) \in \mathbb{R}^{\frac{d}{c}}$, and we concatenate all these outputs to obtain $\boldsymbol{N}_t$ as $\boldsymbol{N}_t = [\boldsymbol{N}_t^1, \ldots, \boldsymbol{N}_t^c]$.

Table 1: Comparison between our approach, GaLore, and LoRA for storing the trainable parameters during fine-tuning with AdamW. For simplicity, assume weight dimensions $d$ can be reshaped to 2D matrix of size $\sqrt{d} \times \sqrt{d}$, $r \ll d$ is the chosen rank, $k \ll d$ is the dimension we want to compress each optimizer state to. The projection matrices refer to the costs of storing $\boldsymbol{B}_t$ during fine-tuning.

|  | MESGC | CESGC | GaLore | LoRA |
| --- | --- | --- | --- | --- |
| Weights | $d$ | $d$ | $d$ | $d + 2r\sqrt{d}$ |
| Optimizer States | $2k$ | $2k$ | $2r\sqrt{d}$ | $4r\sqrt{d}$ |
| Projection Matrices | - | $r\sqrt{d}$ | $r\sqrt{d}$ | - |

We select $s_c = s/c$ non-zero elements per chunk to ensure $s$ non-zero entries overall. Since the projection matrix $\boldsymbol{A}$ is the same for each chunk, we obtain efficiency by a factor of $c$ for storing $\boldsymbol{A}$. However, we may not achieve an exact estimate of $\tilde{\boldsymbol{G}}_t$ and $\tilde{\boldsymbol{G}}_t^2$ when sparsifying and concatenating $\boldsymbol{G}_t^i$'s because the sparsity pattern in $\boldsymbol{G}_t$ is not truly uniform. This performance loss would be more severe with increasing $c$, while it enhances efficiency by reducing the dimension of the projection matrix $\boldsymbol{A}$. We note that the chunking technique introduces more flexibility with the proposed SGC approach in realizing a more diverse spectrum of performance-efficiency tradeoff.

**Compute Efficient SGC (CESGC).** The main tradeoff for our memory efficient approach is increased runtime attributed to OMP, which scales with $d$, the size of gradients $\boldsymbol{G}_t$. Here, we present a computationally efficient alternative at the expense of slightly increased memory usage. For ease of presentation here, consider $\boldsymbol{G}_t \in \mathbb{R}^{m \times n}$ to be in a matrix form. The main idea is to perform double compression, where we first compress $\boldsymbol{G}_t$ once using a projection matrix $\boldsymbol{B}_t \in \mathbb{R}^{r \times m}$, and then apply SGC to this compressed gradient of dimension $(r \times n) \ll d$, therefore reducing time complexity. The intuition behind this approach is that the resultant vector after the first compression is still quasi-sparse. The projection matrix $\boldsymbol{B}_t$ should be selected such that as much information is retained after projection. For this purpose, we use the fact that SGC is orthogonal to many PEFT methods. Thus, we apply one of these methods, GaLore, to obtain $\boldsymbol{B}_t$, which reduces the dimension of the vector entering the SGC algorithm. Specifically, we initialize the projection matrix $\boldsymbol{B}_t$ every fixed number of iterations by applying truncated SVD on $\boldsymbol{G}_t$:

$$\boldsymbol{U}, \Lambda, \boldsymbol{V} = \text{SVD}(\boldsymbol{G}_t), \quad \boldsymbol{B}_t = \boldsymbol{U}[:, :r] \in \mathbb{R}^{r \times m},$$

where $\boldsymbol{B}_t$ is set to be the first $r$ columns of the left-singular vectors of SVD of $\boldsymbol{G}_t$. We then project the gradients $\boldsymbol{G}_t$ using $\boldsymbol{B}_t$ and apply SGC to the resultant vector, i.e., $\text{SGC}(\boldsymbol{B}_t \boldsymbol{G}_t)$. Finally, we project back the resultant updates from $\text{SGC}(\boldsymbol{B}_t \boldsymbol{G}_t)$ onto the original $d$-dimensional space using $\boldsymbol{B}_t^T$ to update the parameters in $\boldsymbol{W}$. Incorporating this into our formulation in 4 yields:

$$\boldsymbol{W}^{(1)} = \boldsymbol{W}^{(0)} - \eta \sum_t \boldsymbol{B}_t^T \text{SGC}(\boldsymbol{B}_t \boldsymbol{G}_t). \tag{15}$$

We note that the dimension of the vector entering SGC is $r \times n$ rather than $d$, thus improving the compute efficiency with OMP. CESGC can be combined with our memory efficient implementation, where chunking is performed after the projection of $\boldsymbol{G}_t$, and we assume this is performed by default for experiments using CESGC. In Appendix C, we discuss some further extensions of SGC.

### 4.5 MEMORY ANALYSIS

Here, we analyze the memory requirements of our efficient SGC implementations and compare it with popular PEFT methods, specifically GaLore and LoRA. The memory requirements of our approach, Galore, and LoRA to perform weight updates for a single vector are shown in Table 1. Observe that the number of optimizer states in both Galore and LoRA are a function of $d$. On the other hand, the size of optimizer states for our memory efficient approach is independent of the weight dimensions, and only depends on $k = \kappa c s_c$, where $s_c$ is sparsity per chunk, $c$ is the number of chunks, and the constant $\kappa$ is to satisfy the RIP conditions for the OMP algorithm. This enables our approach to be significantly more memory efficient in the optimizer states.

To ensure a fair comparison, we analyze the total memory consumption for optimizer states across the entire model and include discussion on how the size of the projection matrix $\boldsymbol{A}$ can impact these numbers. For the following analysis, we will be considering the memory efficient approach

Table 2: LLaMA2-7B, LLaMA3-8B, and LLaMA2-13B on fine-tuning eight commonsense benchmarks (5 shots) using various PEFT methods. Average accuracy is reported in the final column. Note that # Params refers to percentage of optimizer states, $M_t$ and $V_t$, relative to full fine-tuning.

| Model | Method | # Params (%) | ARC-e | ARC-c | BoolQ | HellaSwag | OBQA | PIQA | SIQA | WinoGrande | Average |
|---|---|---|---|---|---|---|---|---|---|---|---|
| LLaMA2-7B | Full Fine-tuning | 100 | 82.5 | 55.4 | 83.8 | 77.8 | 45.8 | 80.1 | 55.4 | 77.8 | 69.8 |
| | CESGC | 0.08 | 82.9 | 53.9 | 82.9 | 77.5 | 44.8 | 79.9 | 54.2 | 74.5 | 68.7 |
| | GaLore | 0.10 | 82.3 | 54.1 | 81.7 | 78.2 | 45.8 | 80.6 | 53.5 | 75.3 | **68.9** |
| | LoRA | 0.20 | 82.1 | 53.2 | 84.3 | 76.2 | 44.0 | 80.4 | 54.0 | 76.5 | 68.8 |
| LLaMA3-8B | Full Fine-tuning | 100 | 85.8 | 62.5 | 86.6 | 81.2 | 51.4 | 82.3 | 59.5 | 81.9 | 73.9 |
| | CESGC | 0.08 | 83.9 | 57.8 | 85.2 | 81.0 | 46.2 | 82.0 | 53.4 | 77.8 | **70.9** |
| | GaLore | 0.10 | 84.3 | 57.2 | 82.6 | 81.2 | 46.2 | 82.3 | 52.9 | 78.0 | 70.6 |
| | LoRA | 0.20 | 82.3 | 56.2 | 83.8 | 79.5 | 48.0 | 81.7 | 52.8 | 74.4 | 69.9 |
| LLaMA2-13B | Full Fine-tuning | 100 | 86.2 | 60.9 | 87.4 | 81.0 | 51.8 | 82.0 | 60.3 | 82.9 | 74.1 |
| | CESGC | 0.07 | 84.1 | 57.2 | 85.3 | 80.0 | 49.4 | 82.0 | 54.6 | 78.6 | 71.4 |
| | GaLore | 0.08 | 83.8 | 56.2 | 85.3 | 81.2 | 47.4 | 81.7 | 55.5 | 79.0 | 71.3 |
| | LoRA | 0.16 | 83.4 | 57.1 | 86.3 | 81.3 | 48.0 | 81.7 | 56.5 | 79.6 | **71.7** |

MESGC only. Assume that we have a model loaded using precision $p$, with $L$ layers and $H$ attention heads, and we are interested in applying PEFT techniques to the query and value attentions each of size $d$. The total memory usage of optimizer states for our approach then becomes $M_{\text{MESGC}} = \frac{\kappa s_c dp}{c} + 4LH\kappa cs_c p$, where we rewrite $k = \kappa cs_c$. The first term of $M_{\text{MESGC}}$ is the memory required to store the projection matrix $\boldsymbol{A} \in \mathbb{R}^{(k \times d)/c}$, and the second term is the total memory requirements for the attention across the entire model.

To demonstrate that our approach can be more memory efficient than any approach that is a function of $d$, assume that there exists a class of solutions $D_{\text{MIN}}$ such that the total number of optimizer states required for AdamW is $2\sqrt{d}$, where we reshape the $d$-dimensional vector to a $\sqrt{d} \times \sqrt{d}$ matrix. This is achieved by setting the rank $r = 1$ for such methods. We present the theoretical minimum for these solutions as $M_{D_{\text{MIN}}} = 4LH\sqrt{d}p$. Finding a set of parameters such that our approach consumes less total memory will require satisfying the inequality of $M_{\text{MESGC}} < M_{D_{\text{MIN}}}$. We will see in the next section that, by assigning values to each of the variables $L, H, \kappa,$ and $d$, there exists a set of solutions for $s_c$ and $c$ such that the inequality is satisfied.

## 5 EXPERIMENTS

We evaluate our approach on fine-tuning languages models using the LLaMA family, specifically on LLaMA2-7B, LLaMA3-8B, and LLaMA2-13B. The results are compared with full fine-tuning, LoRA, and GaLore as baseline for all the setups. In addition, we demonstrate the memory savings of the proposed MESGC algorithm and perform a set of ablation studies to illustrate the tradeoff between chunk size $c$ and sparsity $s$. The results demonstrate how our approach can enable more granular control over the number of optimizer states and achieve comparable accuracy to LoRA and GaLore while using a significantly smaller number of optimizer states.

### 5.1 COMMONSENSE REASONING

We evaluate LLaMA2-7B, LLaMA3-8B, and LLaMA2-13B on a set of commonsense reasoning tasks to demonstrate CESGC's effectiveness in fine-tuning. Commonsense reasoning tasks involve 8 subtasks and we follow Hu et al. (2023) to combine the training sets into a single dataset and evaluate on each of the individual tasks separately. Details of hyperparameters and training settings can be found in Appendix D.1. Results from Table 2 show that our approach achieves a comparable average accuracy compared to both GaLore and LoRA, while using a smaller number of optimizer state parameters. Notably, in the LLaMA3-8B model, CESGC performs the best, achieving a superior accuracy of $1\%$ over LoRA, while using less than half the number of optimizer state parameters.

### 5.2 MEMORY EFFICIENCY

We evaluate the memory efficiency of MESGC when fine-tuning a LLaMA2-7B model. Substituting values into the inequality $M_{\text{MESGC}} < M_{D_{\text{MIN}}}$ with $d = 4096^2, L = 32, H = 32, \kappa = 7$, and rearranging we get:

$$\frac{7c^2 s_c}{4096} + 7s_c < c \quad \Rightarrow \quad s_c < \frac{4096c}{7(c^2 + 4096)}.$$

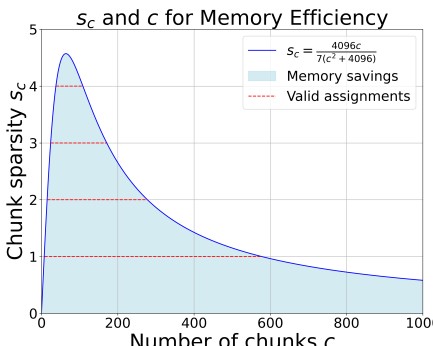

Figure 2: MESGC is more memory efficient in regions highlighted in blue compared to $D_{\text{MIN}}$. The red dotted lines show the valid assignments for positive integers $s_c$.

Table 3: MESGC achieves superior average accuracy when fine-tuning LLaMA2-7B on common-sense reasoning while pushing towards a minimum number of optimizer states. MESGC conducted with $c = 256$, $s_c = 1$, $\kappa = 8$, while both GaLore and LoRA use rank $r = 1$.

| Method | # Params | Accuracy |
|--------|----------|----------|
| MESGC  | 4096     | 68.0     |
| GaLore | 8192     | 67.4     |
| LoRA   | 16384    | 67.7     |

Figure 2 shows the region such that this inequality is satisfied, and we can select a valid combination of $s_c$ and $c$ to achieve lower memory usage than any other PEFT approach dependent on $d$.

To find the minimum memory usage of $M_{\text{MESGC}}$, set $s_c = 1$, and find the critical point by calculating $c$ for $\partial M_{\text{MESGC}}/\partial c = 0$:

$$\frac{-\kappa d}{c^2} + 4LH\kappa c = 0 \quad \Rightarrow \quad c = 64.$$

Consider $r = 1$, the minimum rank used for GaLore and LoRA. Based on Table 1, we can calculate that GaLore and LoRA require 8192 and 16384 optimizer states, respectively. With $s_c = 1$, $c = 64$, and $\kappa = 7$, MESGC requires only 896 optimizer states, reducing the number of parameters by around 10 times. To demonstrate how MESGC performs using a significantly lower number of optimizer states, we fine-tune LLaMA2-7B on a subset of the commonsense reasoning dataset, setting $k = 2048$ (see Appendix D.2 for details). Table 3 shows that our approach achieves $0.6\%$ higher average accuracy than GaLore while using only half the number of optimizer states.

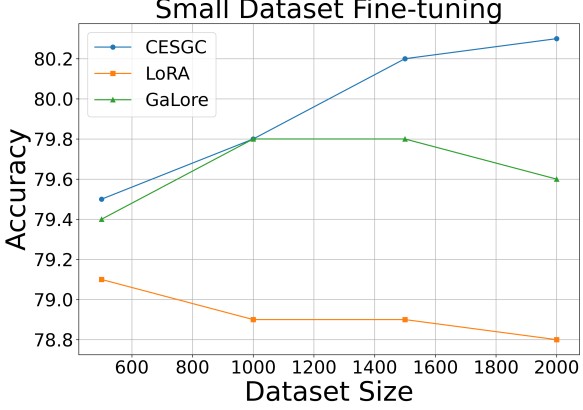

Figure 3: CESGC outperforms both GaLore and LoRA when fine-tuning with limited data on boolQ.

### 5.3 FINE-TUNING ON SMALL DATASETS

To evaluate our approach's effectiveness on small datasets, we focus on fine-tuning LLaMA2-7B on subsets of the boolQ (Clark et al., 2019) dataset while using a minimal number of optimizer states. Specifically, we split the full dataset into multiple subsets ranging from 500 to 2000 samples, and use an equal number of optimizer states across all methods (further details can be found in Appendix D.3). From Figure 3, it can be seen that CESGC performs strictly better using small dataset sizes. We observe that this may be task dependent, but for tasks such as boolQ that rely on leveraging the pre-trained knowledge about facts and entities, our approach can provide a more targeted method for

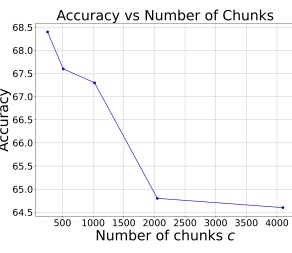 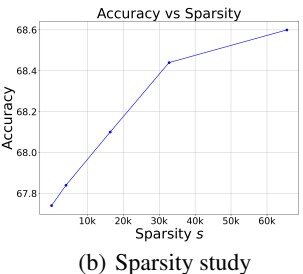 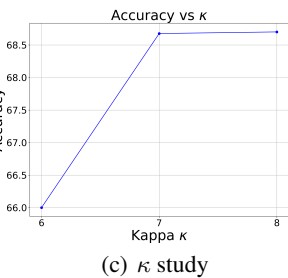

(a) Number of chunks study     (b) Sparsity study     (c) $\kappa$ study

Figure 4: Ablation study for effects of number of chunks $c$, sparsity $s$, and constant $\kappa$. **(a)**. Average accuracy with varying $c$ and constant $s$. **(b)**. Average accuracy with varying $s$ and constant $c$. **(c)**. Average accuracy with varying $\kappa$.

fine-tuning by greedily adjusting based on largest gradient magnitudes. On the other hand, LoRA at the lowest rank ($r = 1$) struggles to learn under the limited dataset scenario, while GaLore with $r = 1$ underperforms CESGC.

## 5.4 ABLATION STUDY

Here, we investigate the effects of number of chunks $c$, total sparsity $s$, and the constant $\kappa$ on fine-tuning performance (details in Appendix D.4). First, we set the total sparsity $s$, to be constant and vary $c$. Figure 4(a) shows that increasing the number of chunks, while keeping the total $s$ constant decreases average accuracy across the commonsense reasoning evaluation. We attribute this to the uniform chunking, where the number of non-zero elements selected per chunk is $s_c = s/c$. However, in practice, the sparsity pattern of gradients may vary across the chunks, with certain parameter regions potentially requiring more attention than others. Therefore, we see higher accuracy corresponding to smaller chunk sizes.

For sparsity, there is a general increasing trend, as seen in Figure 4(b). As the number of non-zero elements selected increases, so does the number of optimizer states $k$, we expect the accuracy to improve until $s$ is equal to the number of parameters, as in full fine-tuning. We observe that increasing $s$ after a certain point results in diminished returns seeing as the slope is most steep when $s$ is increased initially and is less steep afterwards. This can be explained by how a small percentage of parameters account for the majority of the gradient norms during fine-tuning, which is supported by the observations in Song et al. (2024).

Finally, we investigate the effect of $\kappa$, the constant to satisfy the RIP condition, with the goal of finding a lower bound such that performance is not negatively affected. Based on Figure 4(c), we see that if $\kappa$ is set to 6, performance drops significantly. However, there is minimal gain from increasing $\kappa$ from 7 to 8, indicating a $\kappa$ value of 7 should be sufficient.

## 6 CONCLUSION

In this work, we proposed a novel fine-tuning method, SGC, that enables flexible and granular control over the number of optimizer states. The key idea, leveraging the sparsity of the gradients, is to compress them through a linear projection onto a subspace of an arbitrary dimension $k$, which is independent of the original parameter dimensions. The updates are performed within this lower-dimensional subspace, and the results are projected back into the original $d$-dimensional space, effectively utilizing the gradient sparsity. This allows SGC to have significantly smaller and more granular number of parameters to train during fine-tuning compared to other PEFT approaches. We also provided two efficient implementations of SGC, MESGC and CESGC, and show through experiments that our approach can achieve comparable accuracy while being more memory efficient than other PEFT methods. Notably, we demonstrated that our approach achieves superior performance in both data-limited and memory-limited settings, achieving higher accuracy than LoRA and GaLore. Our approach is orthogonal to many gradient compression methods, opening opportunities for future work to integrate them and explore SGC's generalizability in domains like vision and audio.

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

## A    GaLore Analysis

Rather than operating on the parameter space, GaLore saves memory by reducing the number of parameters in the optimizer states (Zhao et al., 2024). Specifically, it projects the gradient $\boldsymbol{G}_t \in \mathbb{R}^{m \times n}$ at each time step $t$ to a lower-dimensional representation $\hat{\boldsymbol{G}}_t = \boldsymbol{P}_t \boldsymbol{G}_t \in R^{r \times n}$ by using a projection matrix $\boldsymbol{P}_t \in \mathbb{R}^{r \times m}$ that is set to the first $r$ columns of the left singular vectors of SVD of $\boldsymbol{G}_t$. The size of the optimizer states, which are equal to the dimensions of the projected gradient $\hat{\boldsymbol{G}}_t$ is then reduced, providing memory savings. However, observe that $\hat{\boldsymbol{G}}_t$ is still dependent on $n$, meaning that, similar to LoRA, there exists a bound dependent on $n$ that we cannot reduce the number of optimizer states any further. Likewise, granularity over parameters is a function of $n$, and tied to the model's weight dimensions.

## B    Efficient Orthogonal Matching Pursuit

Our implementation of OMP is based on the inverse Cholesky factorization method (Zhu et al., 2020), see Algorithm 3. We perform pre-calculation of the gram matrix $\boldsymbol{G}$, to reduce computational costs, but at the same time introduce additional implementation memory requirements. For pure memory efficiency, $\boldsymbol{G}$ should not be pre-computed or alternatively, it is possible to implement a more memory efficient Algorithm 3 at the expense of additional runtime.

---

**Algorithm 3** OMP by Inverse Cholesky Factorization

---

**Inputs:** measurements $\boldsymbol{y}$, projection matrix $\boldsymbol{A}$, sparsity value $s$

**Initialize:** $\Lambda_0 = \emptyset$, the residual $\boldsymbol{r}^{(0)} = \boldsymbol{y}$, gram matrix $\boldsymbol{G} = \boldsymbol{A}^H \boldsymbol{A}$, and the iteration counter $k = 1$.

**while** $k \leq s$ **do**

   **Projection:** if $k = 1$, compute $\boldsymbol{p}^0 = \boldsymbol{A}^H \boldsymbol{r}^0$, else

$$\boldsymbol{p}^{(k-1)} = \boldsymbol{p}^{(k-2)} - \boldsymbol{b}_{:(k-1)} a_{k-1},$$

where $\boldsymbol{b}_{:(k-1)}$ is the $(k-1)$-th column of $\boldsymbol{B}_{k-1}$, and $\boldsymbol{a}_{k-1}$ is the $(k-1)$-th entry of $\boldsymbol{a}_{k-1}$.

   **Select** $i^{(k)} = \arg\max_{i=1,2,\ldots,d} \left( \frac{|p_i^{(k-1)}|}{\|\boldsymbol{A}_{:i}\|} \right)$, where $p_i^{(k-1)}$ is the $i$-th entry of $\boldsymbol{p}^{(k-1)}$.

$$\text{Let } \Lambda_k = \Lambda_{k-1} \cup \{i^{(k)}\}, \quad \text{i.e.,} \quad \lambda_k = i^{(k)} \text{ is the k-th entry of the set } \Lambda_k.$$

   **Obtain**

$$\boldsymbol{c}_{k-1} = \left( \boldsymbol{b}_{\lambda_k, 1:\Lambda_{k-1}}^H \right)^H,$$

where $\boldsymbol{b}_{\lambda_k, 1:\Lambda_{k-1}}$ is the $\lambda_k$-th row of $\boldsymbol{B}_{k-1}$. Then compute $\gamma_k = \frac{1}{\sqrt{g_{\lambda_k, \lambda_k} - \boldsymbol{c}_{k-1}^H \boldsymbol{c}_{k-1}}}$,

$$a_k = \gamma_k p_{\lambda_k}^{k-1},$$

$$\boldsymbol{a}_k = \begin{bmatrix} \boldsymbol{a}_{k-1}^T & a_{:k} \end{bmatrix}^T,$$

$$\boldsymbol{b}_{:k} = \gamma_k \left( \boldsymbol{g}_{:\lambda_k} - \boldsymbol{B}_{k-1} \boldsymbol{c}_{k-1} \right),$$

$$\boldsymbol{B}_k = \begin{bmatrix} \boldsymbol{B}_{k-1}^T & \boldsymbol{b}_{:k} \end{bmatrix},$$

where $p_{\lambda_k}^{k-1}$ is the $\lambda_k$-th entry of $\boldsymbol{p}^{k-1}$, $\boldsymbol{g}_{:\lambda_k}^k$ is the $\lambda_k$-th column of $\boldsymbol{G}$, and $\boldsymbol{c}_0 = \boldsymbol{B}_0 = \boldsymbol{a}_0 = \emptyset$ is assumed for $k = 1$. Finally, if $k = 1$, compute $\boldsymbol{F}_1 = \sqrt{g_{\lambda_1, \lambda_1}}$, else

$$\boldsymbol{F}_k = \begin{bmatrix} \boldsymbol{F}_{k-1} & -\gamma_k \boldsymbol{F}_{k-1} \boldsymbol{c}_{k-1} \\ \boldsymbol{0}_{k-1} & \gamma_k \end{bmatrix},$$

   $k := k + 1$.

**end while**

**Output:** Compute $\hat{\boldsymbol{x}}_s = \boldsymbol{F}_s \boldsymbol{a}_s$, $\boldsymbol{r}^{(s)} = y - \boldsymbol{A}_{\Lambda_s} \hat{\boldsymbol{x}}_s$, and return $\boldsymbol{r}^{(s)}, \Lambda_s, \hat{\boldsymbol{x}}_s$.

---

## C  Extensions of SGC

In practice, having a static projection matrix $\boldsymbol{A}$ is heavily dependent on the initialization, and can potentially lead to slower convergence. To address this, we can adjust $\boldsymbol{A}$ every $T$ iterations, and modify SGC to obtain SGCA outlined in Algorithm 4. Lines 9 initializes a new random projection matrix $\boldsymbol{A}'$ to enable future gradients $\boldsymbol{G}_t$ to be projected into another subspace. Lines $10 - 11$ are necessary to ensure the current $\boldsymbol{M}_t$ and $\boldsymbol{V}_t$ terms are re-aligned using $\boldsymbol{A}'$ such that we can perform OMP at the next time step. Algorithm 4 can improve performance but comes at a cost of increased runtime, since we need to run OMP two more times. Alternatively, it can be possible to store the results from first call but requires additional memory requirements.

---

**Algorithm 4** SGCA at timestep $t$

---

1: **Inputs:** $\boldsymbol{G}_t, \boldsymbol{A}, s, \beta_1, \beta_2, \epsilon$
2: $\boldsymbol{p}_t = \boldsymbol{A} \, \mathrm{Sparsify}_s(\boldsymbol{G}_t), \quad \boldsymbol{q}_t = \boldsymbol{A} \, \mathrm{Sparsify}_s(\boldsymbol{G}_t^2)$
3: $\boldsymbol{M}_t \leftarrow \beta_1 \boldsymbol{M}_{t-1} + (1 - \beta_1)\boldsymbol{p}_t$
4: $\boldsymbol{V}_t \leftarrow \beta_2 \boldsymbol{V}_{t-1} + (1 - \beta_2)\boldsymbol{q}_t$
5: $\boldsymbol{M}_t \leftarrow \frac{\boldsymbol{M}_t}{1 - \beta_1^t}$
6: $\boldsymbol{V}_t \leftarrow \frac{\boldsymbol{V}_t}{1 - \beta_2^t}$
7: $\boldsymbol{N}_t = \alpha \frac{\mathrm{OMP}_{\boldsymbol{A}}(\boldsymbol{M}_t)}{\sqrt{\mathrm{OMP}_{\boldsymbol{A}}(\boldsymbol{V}_t)} + \epsilon}$
8: **if** $t \bmod T = 0$ **then**
9:    $\boldsymbol{A}' \sim \mathcal{N}\left(\boldsymbol{0}, \frac{1}{\sqrt{k}}\boldsymbol{1}\right)$
10:    $\boldsymbol{M}_t = \boldsymbol{A}' \mathrm{OMP}_{\boldsymbol{A}}(\boldsymbol{M}_t)$
11:    $\boldsymbol{V}_t = \boldsymbol{A}' \mathrm{OMP}_{\boldsymbol{A}}(\boldsymbol{V}_t)$
12:    $\boldsymbol{A} = \boldsymbol{A}'$
13: **end if**
14: **return** $\boldsymbol{N}_t$

---

## D  Fine-Tuning Experiments

### D.1  Commonsense Reasoning

We fine-tune pretrained LLaMA2-7B, LLaMA2-13B, and LLaMA3-8B models obtained from Hugging Face. We trained each model for 1 epoch on the full commonsense dataset consisting of 170k examples. For consistency, we used a batch size of 16 across all experiments and train for 1 epoch. Since the goal is to observe performance improvements with only training a limited number of parameters, we only fine-tune on two of the attention matrices, keeping everything else frozen. For LlaMA2-7B and LLaMA-2-13B, we target the query and value matrices, whilst for LLaMA3-8B, we targeted the query output matrices. For LLaMA3-8B, we select the output matrix instead of the value matrix to keep the dimensions consistent for comparison. Full details of hyperparameters can be found in Table 4.

### D.2  Memory Efficiency

For this experiment, we apply the MESGC algorithm. First, we select a subset of 10k examples from the full commonsense dataset and fine-tune the LLaMA2-7B model, evaluating on all commonsense reasoning tasks. We used a batch size of 16 across all experiments and train for 1 epoch is used. The full results can be found in Table 5 and hyperparameters in Table 6.

### D.3  Fine-tuning on Small Datasets

We first obtain a subset consisting of 2000 samples from the boolQ dataset. We then create four partitions of data ranging in size from 500 to 2000 examples, in increments of 500. For this experiment, we are interested in comparing performance between our approach and baselines given equal optimizer state sizes. Thus, we set the total number of optimizer states to 8192, and perform fine-tuning with batch size 16 over 2 epochs using LLaMA2-7B based on the settings shown in Table 7.

Table 4: Hyperparameters used for commonsense reasoning experiments.

| Model | Method | learning rate | rank $r$ | num. chunks $c$ | sparsity $s$ | $\kappa$ | $\alpha$ |
|-------|--------|---------------|----------|-----------------|--------------|----------|----------|
| LLaMA2-7B | Full Finetuning | 1e-5 | - | - | - | - | - |
| | CESGC | 2e-5 | 32 | 64 | 1984 | 7 | 2 |
| | GaLore | 2e-5 | 4 | - | - | - | 2 |
| | LoRA | 1e-4 | 4 | - | - | - | - |
| LLaMA3-8B | Full Finetuning | 1e-5 | - | - | - | - | - |
| | CESGC | 2e-5 | 32 | 64 | 1984 | 7 | 2 |
| | GaLore | 2e-5 | 4 | - | - | - | - |
| | LoRA | 1e-4 | 4 | - | - | - | - |
| LLaMA2-13B | Full Finetuning | 1e-5 | - | - | - | - | - |
| | CESGC | 3e-5 | 32 | 64 | 2496 | 7 | 2 |
| | GaLore | 3e-5 | 4 | - | - | - | 2 |
| | LoRA | 1e-4 | 4 | - | - | - | - |

Table 5: LLaMA2-7B results on commonsense reasoning for MESGC.

| Method | ARC-e | ARC-c | BoolQ | HellaSwag | OBQA | PIQA | SIQA | WinoGrande | Average |
|--------|-------|-------|-------|-----------|------|------|------|------------|---------|
| CESGC | 80.9 | 53.4 | 82.4 | 78.4 | 43.8 | 79.9 | 52.3 | 73.2 | **68.0** |
| GaLore | 80.2 | 52.2 | 79.0 | 78.4 | 43.0 | 80.5 | 51.6 | 74.0 | 67.4 |
| LoRA | 80.9 | 52.2 | 79.5 | 78.5 | 44.6 | 80.0 | 51.7 | 73.9 | 67.7 |

Table 6: Hyperparameters used for commonsense reasoning for MESGC.

| Method | learning rate | rank $r$ | num. chunks $c$ | sparsity $s$ | $\kappa$ | $\alpha$ |
|--------|---------------|----------|-----------------|--------------|----------|----------|
| MESGC | 2e-5 | - | 256 | 256 | 8 | 2 |
| GaLore | 2e-5 | 1 | - | - | - | 2 |
| LoRA | 1e-4 | 1 | - | - | - | - |

Table 7: Hyperparameters used for fine-tuning boolQ.

| Method | learning rate | rank $r$ | num. chunks $c$ | sparsity $s$ | $\kappa$ | $\alpha$ |
|--------|---------------|----------|-----------------|--------------|----------|----------|
| CESGC | 2e-5 | 8 | 64 | 64 | 8 | 2 |
| GaLore | 2e-5 | 1 | - | - | - | 2 |
| LoRA | 1e-4 | 1 | - | - | - | - |

Table 8: Hyperparameters used for ablation study.

| Study | Method | rank $r$ | num. chunks $c$ | sparsity $s$ | $\kappa$ |
|-------|--------|----------|-----------------|--------------|----------|
| Chunks $c$ | MESGC | - | 256, 512, 1024, 2048, 4096 | 4096 | 7 |
| Sparsity $s$ | CESGC | 32 | 64 | 64, 4096, 16384, 32768, 65536 | 7 |
| Kappa $\kappa$ | CESGC | 32 | 64 | 1984 | 6, 7, 8 |

## D.4 ABLATION STUDY

For chunks $c$ and sparsity $s$ studies, we fine-tuned on the LLaMA2-7B model fine-tuned on a subset of 30k examples using commonsense reasoning dataset. For the chunk size study, we performed the experiment based on our MESGC approach, while for sparsity, we used CESGC. Finally, different values of $\kappa$ was tested on the full commonsense dataset using CESGC. The same batch size of 16, training epochs of 1, learning rate, $\eta = 2e^{-5}$ and alpha, $\alpha = 2$ is used for all three studies. Other hyperparameter details are shown in Table 8.

