# OpenReview forum: "Sparse Gradient Compression for Fine-Tuning Large Language Models"
_ICLR.cc/2025/Conference — ICLR 2025 Conference Withdrawn Submission_

### Official Review · Reviewer_W8Gr · 2024-11-02

**Soundness:** 2
**Presentation:** 3
**Contribution:** 2
**Rating:** 5
**Confidence:** 5

**Summary:**

The main contribution of the paper is the introduction of the Sparse Gradient Compression (SGC) method for memory-efficient fine-tuning of large language models (LLMs). SGC leverages inherent sparsity in gradients to compress optimizer states by projecting them onto a lower-dimensional subspace, independent of the model's original parameter dimensions. This approach offers a trade-off between memory efficiency and performance.

**Strengths:**

- The presentation is clear and easy to follow, with only a few minor typos.
- The proposed sparse gradient method is straightforward and supported by reasonable theoretical foundations.

**Weaknesses:**

- **Dataset Limitation**: The authors only use a single dataset (Commonsense) in the experimental sections. I strongly recommend adding at least one more dataset to demonstrate the generalizability of the algorithm across different data domains.

- **Comparison to LoRA in Speed**: While SGC effectively reduces optimizer memory costs similar to LoRA, LoRA offers additional advantages by significantly speeding up the fine-tuning process. Through low-rank adapters and fewer trainable parameters, LoRA can accelerate fine-tuning by 12–16 times compared to full fine-tuning. Since SGC performs full forward and backward propagation, it does not offer the same speed benefit and is likely to be significantly slower than LoRA. I suggest the authors include a training time profile for SGC to clarify this difference.

- **Comparison to LoRA in optimizer Memory**: LoRA also has the advantage of substantially reducing optimizer memory costs. For example, in LLaMA-2-7B full fine-tuning, the optimizer memory cost for a batch size of 128 can reach up to 40GB. With low-rank adapters, this can be reduced to under 1GB. Since SGC performs full forward and backward propagation, it does not reduce optimizer memory cost and is expected to be comparable to full fine-tuning. I recommend the authors discuss SGC's optimizer memory usage in more detail.

- **Base Model Limitations**: Although the authors utilize LLaMA-2-7B, LLaMA-2-13B, and LLaMA-3-8B, I recommend including more state-of-the-art models, such as LLaMA-3.1. Additionally, incorporating models outside the LLaMA family, like Phi-2/Phi-3 or Mistral, could further demonstrate SGC's generalizability across different model architectures.

**Questions:**

- Please refer to some questions I listed in the weakness section.
- Here are a few typos and minor improvements I found in the paper:
1. Abstract: “dimensionality independent of the original model’s parameters” - it might be clearer to specify “independent of the dimensionality of the original model’s parameters.”
2. Introduction (line 23): "exising PEFT methods" should be "existing PEFT methods."
3. Section 2 (line 127): "Adapter-based methods... However, these approaches can introduce latency during inference." - The phrase "these approaches can introduce latency" might read more smoothly as "these methods may increase latency."
4. Equation in Section 3: Ensure consistency with spaces around equations and symbols, especially around parentheses and operators.
5. Section 4.1 (line 217): "sparisfys(·)" should be "sparsifys(·)."
6. Section 4.3 (line 265): "compressed from pt and qt" should read more clearly as "compressed forms, pt and qt."
7. Section 5.3: "boolQ" should consistently be capitalized as "BoolQ" to match standard dataset naming conventions.
8. Section 5.4: "As indicated in 3" should be "As indicated in Equation 3."

- **Time Profiling**: SGC introduces additional computational overhead and extra time costs compared to full fine-tuning. Could the authors provide further discussion on the complexity of this additional computation, along with some time profiling results to illustrate the impact?


I would like to discuss the questions I raised regarding the weaknesses and concerns with the authors. If my concerns are adequately addressed, I would be willing to reconsider my rating.

---

> ### Author Response · Authors · 2024-11-24
> **Response to Reviewer W8Gr (Part 1)**
>
> We sincerely appreciate the time and effort you have devoted to reviewing our work and offering such valuable feedback. Below, we have addressed each of your points in detail. Note that we have split our response into two parts due to the word limit.
>
> ## Weaknesses
> > Dataset Limitation: The authors only use a single dataset (Commonsense) in the experimental sections. I strongly recommend adding at least one more dataset to demonstrate the generalizability of the algorithm across different data domains.
>
> We have added results from a second dataset, alpaca_gpt4, fine-tuned on LLaMA2-7B, and evaluated on the MMLU benchmark. Our results show that under similar memory consumption, our approach is competitive with LoRA and GaLore.
>
> | Method | STEM | Social Science | Humanities | Other | Average |
> |-----------|-----------|-----------|-----------|-----------|-----------|
> | Full fine-tuning| 35.2 | 51.5 | 41.3 | 52 | 44.7|
> | CESGC | 36.1 | 50.4 | 41.3 | 51.4 | 44.5 |
> | GaLore | 35.2 | 50.5| 41.2 | 51.2 | 44.3 |
> | LoRA | 35.3 | 50.8 | 41.8 | 51.2 | 44.5 |
>
> > Comparison to LoRA in Speed: While SGC effectively reduces optimizer memory costs similar to LoRA, LoRA offers additional advantages by significantly speeding up the fine-tuning process. Through low-rank adapters and fewer trainable parameters, LoRA can accelerate fine-tuning by 12–16 times compared to full fine-tuning. Since SGC performs full forward and backward propagation, it does not offer the same speed benefit and is likely to be significantly slower than LoRA. I suggest the authors include a training time profile for SGC to clarify this difference.
>
> The forward propagation speed is the same for all approaches since the same calculations are made (assuming merged LoRA weights, otherwise there is a marginal increase in computation time for LoRA). We would like to emphasize that SGC does not perform full backward propagation, and only updates the target weight matrices as with LoRA and GaLore. SGC does require computing full gradients which is slower than LoRA, but GaLore suffers from the same issue (as seen in table below). The remaining gap in runtime between CESGC and LoRA can be attributed to the OMP calculation, which we will discuss in more detail in your question at the end.
>
> | Method | Runtime (hours) |
> |-----------|-----------|
> | Full Fine-tuning | 4.5 |
> | LoRA | 4.0 |
> | GaLore | 5.0 |
> | CESGC | 7.5 |
>
> > Comparison to LoRA in Activation Memory: LoRA also has the advantage of substantially reducing activation memory costs. For example, in LLaMA-2-7B full fine-tuning, the activation memory cost for a batch size of 128 can reach up to 40GB. With low-rank adapters, this can be reduced to under 1GB. Since SGC performs full forward and backward propagation, it does not reduce activation memory cost and is expected to be comparable to full fine-tuning. I recommend the authors discuss SGC's activation memory usage in more detail.
>
> We would like to clarify that the activation memory usage of LoRA and SGC is identical when applied to the same target layers (e.g., the Q and V weight matrices). If activation checkpointing is not used, the activations for these layers are stored during forward propagation regardless of the method employed. During backpropagation, the computation of gradients with respect to the Q and V matrices requires the same set of activations for both LoRA and SGC. Therefore, the activation memory cost of SGC is not comparable to full fine-tuning, as it is restricted to the activations of the target layers, just like in LoRA.
>
> > Base Model Limitations: Although the authors utilize LLaMA-2-7B, LLaMA-2-13B, and LLaMA-3-8B, I recommend including more state-of-the-art models, such as LLaMA-3.1. Additionally, incorporating models outside the LLaMA family, like Phi-2/Phi-3 or Mistral, could further demonstrate SGC's generalizability across different model architectures.
>
> To extend our experimental results, we fine-tuned LLaMA-3.1-8B on the commonsense dataset. In addition, we also fine-tuned Mistral-7B on a subset of the alpaca gpt4 dataset evaluated on MMLU. With these results, we show that SGC does generalize across different model architectures.
>
> LLaMA3-8.1B
> | Method | Arc-e | Arc-c | BoolQ| HellaSwag| OBQA | PIQA | SIQA | WinoGrande | Average |
> |-----------|-----------|-----------|-----------|-----------|-----------|-----------|-----------|-----------|-----------|
> | CESGC | 56.6 | 83.2 | 85.0 | 79.0 | 47.8 | 81.8 | 54.7 | 76.7 | 70.6 |
> | GaLore | 58.1 | 84.7 | 82.7 | 81.0 | 45.6 | 82.6 | 52.7 | 77.5 | 70.6 |
> | LoRA | 54.7 | 83.5 | 87.2 | 79.6 | 45.0 | 82.7 | 53.7 | 79.5 | 70.7 |
>
> Mistral-7B
> | Method | STEM | Social Science | Humanities | Other | Average |
> |-----------|-----------|-----------|-----------|-----------|-----------|
> | CESGC | 52.3 | 72.6 | 55.9 | 69.2 | 61.84 |
> | GaLore | 52.3 | 72.6 | 56.0 | 69.0 | 61.83 |
> | LoRA | 52.1 | 72.8 | 55.9 | 68.9 | 61.79 |

---

> > ### Author Response · Authors · 2024-11-24
> > **Response to Reviewer W8Gr (Part 2)**
> >
> > ## Questions
> > > SGC introduces additional computational overhead and extra time costs compared to full fine-tuning. Could the authors provide further discussion on the complexity of this additional computation, along with some time profiling results to illustrate the impact?
> >
> > The extra time complexity of SGC mainly comes from the OMP calculation which requires an extra O(k*s) at each timestep with the efficient batched GPU implementation. To mitigate this, we implemented a GPU batch efficient implementation of OMP that reduces the runtime by over 10 times compared to a standard implementation that does not utilize the GPU. We then provided a compute efficient approach CESGC that further reduces runtime such that it is much closer to other existing approaches. We note that there are still further optimizations that can be made to the OMP signal recovery algorithm and leave that as part of future work. Time profiling results can be found in the table above.

---

### Official Review · Reviewer_5vGv · 2024-11-02

**Soundness:** 3
**Presentation:** 3
**Contribution:** 2
**Rating:** 5
**Confidence:** 3

**Summary:**

This paper introduces Sparse Gradient Compression (SGC), a method aimed at reducing memory requirements when fine-tuning LLMs. SGC leverages gradient sparsity to update optimizer states within a low-dimensional subspace, effectively reducing memory usage while preserving performance. Experimental results demonstrate that SGC outperforms traditional parameter-efficient fine-tuning methods, especially in memory-limited settings.

**Strengths:**

1. The paper presents a novel approach that addresses memory efficiency in large-scale fine-tuning tasks. The proposed approach enables more flexible and granular control over the number of parameters to train during finetuning.
2. Experimental evaluation shows that SGC competes well with and often outperforms existing methods (e.g., LoRA, GaLore) in terms of memory efficiency and accuracy.

**Weaknesses:**

1. The author highlights limitations in the flexibility and granularity of LoRA due to the dependency on model dimensions. However, in practical applications, these constraints may not significantly impact performance. Many real-world tasks do not require extreme reductions in trainable parameters, and the existing flexibility of LoRA is often sufficient. For instance, as shown in Table 2, LoRA fine-tunes only 0.2% of the parameters, meaning the LoRA weights and optimizer states are not the bottleneck—the base model weights and activations are. Reducing this further to 0.08% would likely not yield significant benefits.

**Questions:**

1. Can you justify in what scenarios is fine-grained control over the training parameters necessary?
2. If fine-grained control of training parameters is required, are there simpler methods to achieve similar results, such as using different ranks in different transformer layers?

---

> ### Author Response · Authors · 2024-11-24
> **Response to Reviewer 5vGv**
>
> We sincerely appreciate the time and effort you have devoted to reviewing our work and offering such valuable feedback. Below, we have addressed each of your points in detail.
>
> ## Weaknesses
> > The author highlights limitations in the flexibility and granularity of LoRA due to the dependency on model dimensions. However, in practical applications, these constraints may not significantly impact performance. Many real-world tasks do not require extreme reductions in trainable parameters, and the existing flexibility of LoRA is often sufficient. For instance, as shown in Table 2, LoRA fine-tunes only 0.2% of the parameters, meaning the LoRA weights and optimizer states are not the bottleneck—the base model weights and activations are. Reducing this further to 0.08% would likely not yield significant benefits.
>
> The importance of flexibility and granularity becomes apparent in limited-data fine-tuning. SGC’s flexibility allows it to adapt to diverse resource constraints such as extreme memory limitations or specific downstream task performance requirements. Here we present a scenario that offers practical performance and efficiency benefits as a result of the enhanced flexibility of SGC. In this example, we fine-tune LLaMA2-7B on a 2k subset of the BoolQ dataset and show how under extreme memory requirements, SGC outperforms both GaLore and LoRA. With LoRA and GaLore, r=1 is the minimum number of optimizer parameters (equivalent to total sparisty = 512 in table below). However, for our approach, we can use a smaller number of optimizer parameters by varying the total sparsity level. By going below r=1, we can select values for total sparsity such that SGC outperforms LoRA and GaLore while using a significantly smaller number of optimizer states.
>
> | Total Sparsity| Accuracy (LoRA) | Accuracy (GaLore) | Accuracy (CESGC) |
> |-----------|-----------|-----------|-----------|
> | 64 | - | - | 79.5 |
> | 128 | - | - | 79.4 |
> | 192 | - | - | 79.5 |
> | 256 | - | - | 79.6 |
> | 320| - |  - | 77.9 |
> | 384 | - | - | 80.2 |
> | 448 | - | - | 80.1 |
> | 512 ~ r=1 | 79.1 | 79.4 | 80.4 |
>
> ## Questions
> > Can you justify in what scenarios is fine-grained control over the training parameters necessary?
>
> Please see the answer to the weakness above.
>
> >  If fine-grained control of training parameters is required, are there simpler methods to achieve similar results, such as using different ranks in different transformer layers?
>
> Alternative methods such as layer-wise rank adjustments are tightly coupled to either the model architecture or downstream task, requiring careful manual tuning of which layers should receive higher or lower ranks. In contrast, SGC provides model-agnostic flexibility by decoupling optimizer state size from parameter dimensions, enabling granular memory-performance tradeoffs without requiring architectural modifications. SGC’s low-dimensional projection operates uniformly across all layers, simplifying implementation and making it more scalable for large-scale models with many layers.

---

> > ### Comment · Reviewer_5vGv · 2024-11-25
> >
> > Thank you for the clarification. However, I'm still not entirely convinced about the significance of these 'extreme memory limitations' that can accommodate only the base model but not the base model with an r=1 adapter. If these limitations are indeed an issue, it might be more reasonable to focus on compressing the base model itself, such as with QLoRA, rather than the lora part.

---

### Official Review · Reviewer_eUW7 · 2024-11-03

**Soundness:** 2
**Presentation:** 3
**Contribution:** 2
**Rating:** 5
**Confidence:** 3

**Summary:**

This paper introduces a new optimizer, Sparse Gradient Compression (SGC), designed to enhance fine-tuning efficiency for large language models (LLMs). The main innovation of SGC lies in compressing the optimizer states, providing a more flexible and granular tradeoff between memory usage and performance compared to other Parameter Efficient Fine-Tuning (PEFT) methods. The compression leverages the inherent sparsity in gradients, with the recovery of compressed states performed through a greedy algorithm called Orthogonal Matching Pursuit (OMP). Experimental results on LLaMA models demonstrate that SGC achieves performance comparable to or even better than existing PEFT methods on some tasks, while reducing memory requirements for optimizer states.

**Strengths:**

1. The writing is clear and well-structured, with an appropriate balance of detail, making it easy to understand. Most technical choices are well-motivated and thoroughly explained.
2. The motivation behind the proposed SGC method is intuitive, making the approach conceptually accessible and logical given the challenges in fine-tuning large language models.
3. The experiments and comparative analyses effectively demonstrate that SGC offers memory savings while maintaining comparable performance, validating the practical advantages of the approach.

**Weaknesses:**

1. **Limited Applicability**: While the paper claims that SGC offers a more flexible, fine-grained tradeoff, PEFT methods typically target compute-constrained scenarios, where such granular control may require extra tuning that reduces practicality. It would be beneficial to include a plot with sparsity on the x-axis and performance on the y-axis to directly compare the flexibility of SGC with LoRA. This visualization could more intuitively demonstrate whether SGC’s fine-grained control offers practical performance benefits at different sparsity levels.
2. **Questionable Memory Advantage**: The memory usage for first-order optimization methods largely comes from the model parameters, gradients, activations, and optimizer states. Even with Adam’s two states, optimizer memory costs are typically less than half. SGC, based on Adam, can’t reduce memory below that of simple SGD without momentum, and since it still calculates full gradients, its GPU memory consumption may surpass LoRA, which doesn’t require full gradient computations.
3. **Subpar Performance**: As seen in Table 2, SGC shows no clear performance advantage over methods like LoRA and GaLore, raising questions about its efficacy as a fine-tuning method.
4. **Lack of Related Work Comparison**: The paper omits discussion and comparison with relevant optimizers like Adafactor[1] and CAME[2], which also focus on compressing optimizer states. These omissions reduce the context for understanding SGC’s place among similar methods. Including a comparison on task performance, memory efficiency and convergence speed would better contextualize SGC's advantages and place among similar methods.

**References**:

[1] Shazeer, N., & Stern, M. (2018, July). Adafactor: Adaptive learning rates with sublinear memory cost. In *International Conference on Machine Learning* (pp. 4596-4604). PMLR.

[2] Luo, Y., Ren, X., Zheng, Z., Jiang, Z., Jiang, X., & You, Y. (2023). CAME: Confidence-guided Adaptive Memory Efficient Optimization. In *Proceedings of the 61st Annual Meeting of the Association for Computational Linguistics (Volume 1: Long Papers)*, pages 4442–4453, Toronto, Canada. Association for Computational Linguistics.

**Questions:**

1. In Equation (6), $\boldsymbol{A}$ is initialized randomly as stated in the appendix. How does this randomness affect the model's final performance? Are there significant differences observed across different random seeds?
2. What are the actual comparisons of wall-clock time per iteration and GPU memory usage when training with SGC compared to LoRA and full Adam fine-tuning? Providing this data for LLaMa-7b in a table or plot format would help clarify SGC's efficiency across contexts.
3. Does SGC offer any advantages over Adafactor and CAME in terms of performance or efficiency? If so, could you elaborate on these advantages?

---

> ### Author Response · Authors · 2024-11-24
> **Response to Reviewer eUW7 (Part 1)**
>
> We sincerely appreciate the time and effort you have devoted to reviewing our work and offering such valuable feedback. Below, we have addressed each of your points in detail. Note that we have split our response into two parts due to the word limit.
>
> ## Weaknesses
> > Limited Applicability: While the paper claims that SGC offers a more flexible, fine-grained tradeoff, PEFT methods typically target compute-constrained scenarios, where such granular control may require extra tuning that reduces practicality. It would be beneficial to include a plot with sparsity on the x-axis and performance on the y-axis to directly compare the flexibility of SGC with LoRA. This visualization could more intuitively demonstrate whether SGC’s fine-grained control offers practical performance benefits at different sparsity levels.
>
> Thank you for your observation regarding the tradeoff between flexibility and practicality in compute-constrained scenarios. Below, we clarify the advantages of this tradeoff and address concerns about practicality.
>
> While PEFT methods focus on compute-constrained scenarios, SGC’s flexibility allows it to adapt to diverse resource constraints (such as extreme memory limitations or specific downstream task performance requirements). The additional hyperparameters introduced by SGC can be optimized through automated techniques such as grid search mitigating the overhead of manual tuning. Here we present a scenario that offers practical performance benefits at different sparsity levels. Since we are unable to include a plot in this response, we present the results in a table format. In this example, we fine-tune LLaMA2-7B on a 2k subset of the BoolQ dataset and show how under extreme memory requirements, SGC outperforms both GaLore and LoRA. With LoRA and GaLore, r=1 is the minimum number of optimizer parameters (equivalent to total sparisty = 512 in table below). However, for our approach, we can use a smaller number of optimizer parameters by varying the total sparsity level.  By going below r=1, we can select values for total sparsity such that SGC outperforms LoRA and GaLore while using a significantly smaller number of optimizer states.
>
> | Total Sparsity| Accuracy (LoRA) | Accuracy (GaLore) | Accuracy (CESGC) |
> |-----------|-----------|-----------|-----------|
> | 64 | - | - | 79.5 |
> | 128 | - | - | 79.4 |
> | 192 | - | - | 79.5 |
> | 256 | - | - | 79.6 |
> | 320| - |  - | 77.9 |
> | 384 | - | - | 80.2 |
> | 448 | - | - | 80.1 |
> | 512 ~ r=1 | 79.1 | 79.4 | 80.4 |
>
> > Questionable Memory Advantage: The memory usage for first-order optimization methods largely comes from the model parameters, gradients, activations, and optimizer states. Even with Adam’s two states, optimizer memory costs are typically less than half. SGC, based on Adam, can’t reduce memory below that of simple SGD without momentum, and since it still calculates full gradients, its GPU memory consumption may surpass LoRA, which doesn’t require full gradient computations.
>
> Thank you for raising this point. Instead of storing all gradients in memory, we use per-layer weight updates [1], which compute the gradients only for the current layer being processed. This approach significantly reduces memory requirements by avoiding the need to retain gradients for the entire model at once. By using a memory profiler, we can show that our approach does not require more GPU memory than LoRA given similar optimizer state sizes (see table below).
>
> | Method | Max Memory (GB) |
> |-----------|-----------|
> | Full Fine-tuning | 62.82 |
> | LoRA | 18.87 |
> | GaLore | 18.86 |
> | CESGC | 18.86 |
>
> > Subpar Performance: As seen in Table 2, SGC shows no clear performance advantage over methods like LoRA and GaLore, raising questions about its efficacy as a fine-tuning method.
>
> We acknowledge that Table 2 does not show a consistent performance advantage for SGC across all metrics, and we appreciate the opportunity to clarify and expand on its contributions.
> Firstly, Table 2 demonstrates that SGC achieves comparable accuracy to LoRA and GaLore while using significantly fewer optimizer state parameters, showing that it is still an effective method for fine-tuning.
> The performance of SGC also becomes evident in memory-limited or data-limited settings, as demonstrated in Sections 5.2 and 5.3. In small datasets, CESGC consistently outperforms LoRA and GaLore (e.g., on BoolQ in Figure 3).
> In addition, MESGC also achieves higher accuracy while being more memory efficient in commonsense reasoning tasks (see Table 3).
>
> [1] Lv, Kai, et al. "Adalomo: Low-memory optimization with adaptive learning rate." arXiv preprint arXiv:2310.10195 (2023).

---

> > ### Author Response · Authors · 2024-11-24
> > **Response to Reviewer eUW7 (Part 2)**
> >
> > > Lack of Related Work Comparison: The paper omits discussion and comparison with relevant optimizers like Adafactor[1] and CAME[2], which also focus on compressing optimizer states. These omissions reduce the context for understanding SGC’s place among similar methods. Including a comparison on task performance, memory efficiency and convergence speed would better contextualize SGC's advantages and place among similar methods.
> >
> > Thank you for highlighting the omission of a discussion with relevant optimizers such as Adafactor [1] and CAME [2], and we will be sure to cite these in the final paper. Adafactor is an optimization method based on Adam that reduces memory usage  by approximating second-moment statistics using a factored representation. CAME further extends Adafactor by introducing a confidence-guided strategy to address the training instability and achieves performance improvement while also being memory efficient. We view SGC as primarily a gradient compression method and is thus orthogonal to both approaches. Therefore, with appropriate modifications (e.g., similar to AdamW), SGC can be integrated with optimizers like Adafactor and CAME to further improve memory efficiency.
> >
> > ## Questions
> >
> > > In Equation (6), A is initialized randomly as stated in the appendix. How does this randomness affect the model's final performance? Are there significant differences observed across different random seeds?
> >
> > The randomness does not significantly affect the model’s final performance, and we did not observe any significant differences across different random seeds. As long as A is defined with a zero mean normal distribution with standard deviation $1/\sqrt{k}$, the restricted isometry property is satisfied such that the OMP algorithm can be used accurately.
> >
> > >  What are the actual comparisons of wall-clock time per iteration and GPU memory usage when training with SGC compared to LoRA and full Adam fine-tuning? Providing this data for LLaMa-7b in a table or plot format would help clarify SGC's efficiency across contexts.
> >
> > We fine-tuned LLaMA2-7B on a subset of the commonsense reasoning dataset and details for the time per iteration can be found below. We acknowledge the extra time complexity of SGC and have made efforts to address this. Specifically, we implemented a GPU batch efficient implementation of OMP that reduces the runtime by over 10 times compared to a standard implementation that does not utilize the GPU. We then provided a compute efficient approach CESGC that further reduces runtime such that it is much closer to other existing approaches.
> >
> > | Method | Wall clock time per iteration (s) |
> > |-----------|-----------|
> > | Full Fine-tuning | 1.69 |
> > | LoRA | 4.0 | 1.51 |
> > | GaLore | 1.88 |
> > | CESGC | 2.82 |
> > | MESGC | 7.52 |
> >
> >
> > > Does SGC offer any advantages over Adafactor and CAME in terms of performance or efficiency? If so, could you elaborate on these advantages?
> >
> > We would like to emphasise that SGC is orthogonal to both approaches and can be integrated for greater overall efficiency with various optimizers such as Adafactor and CAME. SGC is a technique for conducting the processing within an optimizer over a subspace with an arbitrary small size. We will make sure to include this discussion in the paper for further clarification.

---

> > > ### Comment · Reviewer_eUW7 · 2024-11-24
> > >
> > > Thank you for your detailed response to my review. After reviewing your reply, I would like to provide the following feedback:
> > >
> > > 1. **On Flexibility:**
> > >    The claimed flexibility of SGC, allowing sparsity levels smaller than LoRA’s $r=1$, is overstated and not well-supported by the evidence provided. Flexibility implies consistent advantages across a wide range of sparsity levels, for example, including $r=4$ and $r=8$. However, your results suggest that SGC only outperforms LoRA under extreme memory constraints where LoRA cannot operate. This is not indicative of true flexibility but rather positions SGC as a niche method for rare scenarios.
> > >
> > >    Furthermore, even in these rare scenarios, the practical benefits are negligible. From the memory usage data you provided, SGC’s advantage over LoRA is less than 0.1% (18.86GB vs. 18.87GB), and this minimal difference is achieved only by exclusively applying the per-layer weight updates strategy to SGC. This undermines the claim of SGC offering meaningful memory efficiency and further suggests that its flexibility is overstated.
> > >
> > > 2. **On Memory Usage:**
> > >    While the per-layer weight updates strategy is effective in reducing memory usage, this is not a novel contribution of SGC. The same strategy can be seamlessly integrated into other baselines, including LoRA and full Adam fine-tuning. Applying this optimization exclusively to SGC introduces a bias in comparisons, especially given that this approach inherently adds computational overhead. A fair evaluation would require applying the same memory-saving techniques to all baselines, which has not been done in this work. As a result, the memory efficiency claims for SGC are not convincing.
> > >
> > >    BTW, why is the use of per-layer weight updates not mentioned in the paper? This is a critical implementation detail, and omitting it from the paper raises questions about the transparency of the comparisons.
> > >
> > > 3. **On Performance Claims:**
> > >    The statement that *“SGC achieves comparable accuracy to LoRA and GaLore while using significantly fewer optimizer state parameters, showing that it is still an effective method for fine-tuning”* is problematic. The reduction in optimizer state parameters does not provide practical benefits if it does not translate to real-world advantages such as improved memory usage or reduced computational overhead. The actual memory analysis and wall-clock time analysis you provided shows no improvement over LoRA, further weakening the practical impact of this claim.
> > >
> > > 4. **On Comparisons with Related Work:**
> > >    The argument that SGC is a gradient compression method and thus orthogonal to memory-efficient optimizers like Adafactor and CAME is not convincing. If differences in technical motivation alone justify excluding Adafactor from comparisons, it raises the question of why SGC is compared to PEFT methods like LoRA, which are fundamentally different from gradient compression approaches. The distinction between SGC and these optimizers lies primarily in whether compression occurs before or after computing optimizer states, and the resulting effects on memory efficiency are highly comparable. Thus, a direct comparison with Adafactor is both reasonable and necessary to establish SGC's relative strengths and weaknesses.

---

### Official Review · Reviewer_7ctj · 2024-11-03

**Soundness:** 2
**Presentation:** 2
**Contribution:** 2
**Rating:** 5
**Confidence:** 4

**Summary:**

Based on gradient sparsity, this paper proposes a flexible gradient compression method to reduce memory usage during training LLMs. By sparsifying and projecting the gradient onto a low-dimensional subspace, the optimizer state is updated, and then, when updating parameters, it is mapped back to the high-dimensional space using the orthogonal matching pursuit (OMP) algorithm.

**Strengths:**

1. SGC is more flexible compared to previous methods like LoRA and GaLore, allowing more granular control over the dimensionality of the compressed optimizer state.
2. On commonsense benchmarks, SGC achieves a comparable average accuracy to both GaLore and LoRA while using fewer optimizer state parameters.

**Weaknesses:**

1. Although SGC is more flexible, this advantage is somewhat marginal, as LoRA and GaLore are already quite flexible.
2. The paper lacks throughput experiments and runtime analysis of OMP.
3. It does not include empirical experiments comparing memory usage of SGC and baseline methods to validate the theoretical analysis.
4. There is no information on the error magnitude after gradient compression.
5. Mischaracterization in lines 350-352 and 368-369, where it mentions GaLore as a type of PEFT method; GaLore is not a PEFT method.

**Questions:**

What is the practical benefit of increased flexibility and granular control, given that memory usage is primarily dominated by parameters rather than optimizer states in methods like PEFT or GaLore?

---

> ### Author Response · Authors · 2024-11-24
> **Response to Reviewer 7ctj**
>
> We sincerely appreciate the time and effort you have devoted to reviewing our work and offering such valuable feedback. Below, we have addressed each of your points in detail.
>
> ## Weaknesses
>
> > Although SGC is more flexible, this advantage is somewhat marginal, as LoRA and GaLore are already quite flexible.
>
> SGC’s flexibility allows it to adapt to diverse resource constraints such as extreme memory limitations or specific downstream task performance requirements. Here we present a scenario that offers practical performance and efficiency benefits as a result of the enhanced flexibility of SGC. In this example, we fine-tune LLaMA2-7B on a 2k subset of the BoolQ dataset and show how under extreme memory requirements, SGC outperforms both GaLore and LoRA. With LoRA and GaLore, r=1 is the minimum number of optimizer parameters (equivalent to total sparisty = 512 in table below). However, SGC can use a smaller number of optimizer parameters by varying the total sparsity level. By going below r=1, we can select values for total sparsity such that SGC outperforms LoRA and GaLore while using a significantly smaller number of optimizer states.
>
> | Total Sparsity| Accuracy (LoRA) | Accuracy (GaLore) | Accuracy (CESGC) |
> |-----------|-----------|-----------|-----------|
> | 64 | - | - | 79.5 |
> | 128 | - | - | 79.4 |
> | 192 | - | - | 79.5 |
> | 256 | - | - | 79.6 |
> | 320| - |  - | 77.9 |
> | 384 | - | - | 80.2 |
> | 448 | - | - | 80.1 |
> | 512 ~ r=1 | 79.1 | 79.4 | 80.4 |
>
> > The paper lacks throughput experiments and runtime analysis of OMP.
>
> We conducted throughput experiment for CESGC fine-tuning LLaMA2-7B on commonsense dataset = 627 tokens/s.
>
> The additional time complexity of our approach mainly comes from the OMP calculation which requires an extra O(k*s) time complexity at each timestep with the efficient batched GPU implementation that we developed. Time profiling results are shown below where we performed a runtime comparison when fine-tuning LLaMA2-7B on the commonsense 170k dataset for one epoch, on a single H100 GPU.
>
> | Method | Runtime (hours) |
> |-----------|-----------|
> | Full Fine-tuning | 4.5 |
> | LoRA | 4.0 |
> | GaLore | 5.0 |
> | CESGC | 7.5 |
>
> > It does not include empirical experiments comparing memory usage of SGC and baseline methods to validate the theoretical analysis.
>
> Empirical experiments for fine-tuning LLaMA2-7b on a subset of the commonsense dataset shows approximately equal max GPU memory usage during the training process. Here memory usage is almost equivalent between LoRA, GaLore, and SGC because parameters $k$ (SGC) and $r$ (LoRA, GaLore) were deliberately selected to be equal. Even in this setting, we can see that SGC is more memory efficient than LoRA, and this gap would increase if we increase the model size and/or the targets to apply SGC to (Only Q, V weight matrices were targeted here).
>
> | Method | Max Memory (GB) |
> |-----------|-----------|
> | Full Fine-tuning | 62.82 |
> | LoRA | 18.87 |
> | GaLore | 18.86 |
> | CESGC | 18.86 |
>
> > There is no information on the error magnitude after gradient compression.
>
> We conducted additional experiments to obtain the error magnitude from before gradient compression, and after OMP recovery. We used the average MSE across all targeted weights to compute the difference between MESGC and a sparsified gradient approach without compression.
> | Model | Average MSE |
> |-----------|-----------|
> | LLaMA2-7B| 0.0191 |
> | LLaMA3-8B| 0.0790 |
> | LLaMA2-13B | 0.0771 |
>
> > Mischaracterization in lines 350-352 and 368-369, where it mentions GaLore as a type of PEFT method; GaLore is not a PEFT method.
>
> Thank you for pointing out this important clarification regarding the characterization of GaLore. We agree  that GaLore is primarily a gradient compression method and not a Parameter-Efficient Fine-Tuning (PEFT) method. We will make sure to fix this in the final version of the paper.
>
> Proposed revisions:
> Lines 350-352: For this purpose, we leverage the compatibility of SGC with various optimization techniques. Specifically, we utilize GaLore, a gradient compression method, to obtain $\mathbf{B}_t$​, which reduces the dimensionality of the vector input to the SGC algorithm.
>
> Lines 368-369: Here, we analyze the memory requirements of our efficient SGC implementations and compare them with established methods for reducing memory usage during fine-tuning, specifically gradient compression techniques like GaLore and PEFT methods like LoRA.
>
> ## Questions
>
> > What is the practical benefit of increased flexibility and granular control, given that memory usage is primarily dominated by parameters rather than optimizer states in methods like PEFT or GaLore?
>
> Please see the answer to the first weakness above.

---

> > ### Comment · Reviewer_7ctj · 2024-11-24
> >
> > Thank you for the feedback. It would be more convincing if the table 1 contained the memory usage corresponding to Total Sparsity.

---

### Official Review · Reviewer_FUbt · 2024-11-03

**Soundness:** 3
**Presentation:** 3
**Contribution:** 2
**Rating:** 5
**Confidence:** 5

**Summary:**

The paper presents a Sparse Gradient Compression (SGC) algorithm to flexibly and granularly control the number of trainable parameters during fine-tuning. By projecting the optimizer states into a subspace, SGC updates and stores these states in a low-dimensional space. Numerical experiments demonstrate that the performance of SGC is comparable to existing parameter-efficient fine-tuning (PEFT) algorithms to some extent.

**Strengths:**

1.The paper introduces two algorithms, MESGC and CESGC, for effectively reducing the memory and computational complexity, respectively.
2.It presents a well-reasoned approach for determining the hyperparameters of the SGC algorithm in Section 5.2.

**Weaknesses:**

1.The novelty of the proposed approach is limited. The concept of projecting optimizer states into a subspace with a dimension independent of the original model size has been previously discussed in the top-k compressor as shown in [1] and [2].
2.The paper lacks a theoretical analysis of the relationship between the choice of k and the model’s convergence, a detail that has been explored in [1] and [2].
3.The idea behind SGC lacks novelty, as both algorithms are quite similar to GaLore. Additionally, MESGC appears to be time-consuming, particularly regarding the practical adjustments outlined in Algorithm 4, and the numerical results in Table 2 do not consistently demonstrate CESGC’s superiority over GaLore.

[1] Stich, S. U., Cordonnier, J. B., & Jaggi, M. (2018). Sparsified SGD with memory. Advances in neural information processing systems, 31.
[2] Li, X., Karimi, B., & Li, P. (2022). On distributed adaptive optimization with gradient compression. arXiv preprint arXiv:2205.05632.

**Questions:**

1.In Section 4.4, the gradient is chunked during computation. If c is not large, the size of the projection matrix remains significant, leading to high memory consumption. Conversely, if c is large, it introduces c times the matrix multiplications, which may be time-consuming. What is the result of the runtime comparison between MESGC and existing PEFT methods?
2.What is the performance of MESGC on small datasets, as discussed in Section 5.3?

---

> ### Author Response · Authors · 2024-11-24
> **Response to Reviewer FUbt (Part 1)**
>
> We sincerely appreciate the time and effort you have devoted to reviewing our work and offering such valuable feedback. Below, we have addressed each of your points in detail. Note that we have split our response into two parts due to the word limit.
>
> ## Weaknesses
>
> > The novelty of the proposed approach is limited. The concept of projecting optimizer states into a subspace with a dimension independent of the original model size has been previously discussed in the top-k compressor as shown in [1] and [2].
>
> Thank you for bringing these papers to our attention and we will be sure to cite them as part of the related work section. Papers [1] and [2] are focused on the distributed setting where communication between workers is the bottleneck and there is an emphasis on error compensation to mitigate the lossy compression. However, the idea of using top-k sparsification along with dimensionality reduction has not been readily explored in the context of fine-tuning LLMs. The gradient compressors in these papers are functions from $\mathbb{R}^d$  to $\mathbb{R}^d$, and the dimensionality of the optimizer states remains unchanged. On the other hand, in our approach, we use a random matrix to project the gradients to a lower dimensional subspace in $\mathbb{R}^k$ (with an arbitrary size $k << d$) while doing sparsification to enable memory efficient storage of optimizer states.
>
> > The paper lacks a theoretical analysis of the relationship between the choice of k and the model’s convergence, a detail that has been explored in [1] and [2].
>
> Following [1], it is possible to show that top-$ k $ sparsification leads to convergence at the same rate as vanilla SGD. The key difference in our algorithm is the use of chunking and sparsification applied to every chunk. Thus, the proof of convergence boils down to bounding the distance between the sparse form of gradient vector $G$ and the sparse form of every sub-vector after chunking the gradient vector $G$. Let us denote $G=[G_1, \dots, G_c]$, where $G_i \in \mathbb{R}^{d/c} $ is the subvector corresponding to each chunk. We then represent the $s$-sparse form of $G$ with $G’$, i.e.,
>
> $$
> G' = \text{Sparsify}_s(G) = \text{Sparsify}_s([G_1, \dots, G_c]),
> $$
>
>
>
>
> and the vector of sparsified chunks with
>
> $$
> \tilde{G} = [\tilde{G}_1 , \dots, \tilde{G}_c], \quad \tilde{G}_i = \text{Sparsify}s_c(G_i).
> $$
>
> In the following, we will provide a bound for $ E \[ \|| G' - \tilde{G} \||_2^2 \]$ where the randomness is from the gradient $G$.
>
> If we assume a uniform distribution for non-zero indices in $ G' $, it is easy to verify that
>
> $$
> E \left[ \|| G' - \tilde{G} \||_2^2 \right] = 0.
> $$
>
> On the other hand, it is straightforward to show that the worst case for $E \[ \|| G' - \tilde{G} \||_2^2 \]$ occurs when all the non-zero entries are located contiguously in $ G' $. This is due to the way we perform the chunking, i.e., having consecutive indices in a chunk. Without loss of generality and for ease of presentation, we assume that entries $ 1 $ to $ s $ in $ G' $ are non-zero.
>
> Let us denote the number of chunks these $ s $ entries span is equal to
>
> $$
> l = \lceil s / (d / c) \rceil.
> $$
>
> Therefore, the error can be computed in two parts: $D_1$, due to the contribution from chunks $ i \leq l $, where there are missing entries not selected in $\tilde{G}$, and secondly, $D_2$, in chunks $ i > l $, where extra entries are selected in $\tilde{G}$. Adding these two together, we get:
>
> $$ E\left[ \|| G' - \tilde{G} \||^2_2 \right] = D_1 + D_2, \text{where}$$
>
> $$ D_1 = E \left[ \sum_{i=1}^l || G_i - \tilde{G}_i ||^2_2 \right] \le (s - l s_c) E \left[ \frac{\|| G' ||^2_2}{s} \right],$$
>
> $$ D_2 = E \left[ \sum_{i=l+1}^c || G_i - \tilde{G}_i ||^2_2 \right] \le (c - l)s_c E\left[ \frac{\|| G' ||^2_2}{s} \right],$$
>
> Putting everything together and simplifying, we get a worst-case bound:
>
> $$
> E\left[ \|| G' - \tilde{G} \||_2^2 \right] \leq 2 \left( 1 - \frac{s}{d} \right) E\left[ \|| G' \||_2^2 \right] \le 2 \left ( 1 - \frac{s}{d} \right) G_m,
> $$
>
> where $G_m$ is the bound over the expectation of the l-2 norm of the gradients.
>
> Using this bound, we can then formulate the theoretical analysis for convergence, and we leave this as future work.

---

> > ### Author Response · Authors · 2024-11-24
> > **Response to Reviewer FUbt (Part 2)**
> >
> > > The idea behind SGC lacks novelty, as both algorithms are quite similar to GaLore.
> >
> > The core idea of SGC differs from GaLore. SGC uses a random matrix $\mathbf{A}$ to project sparsified gradients into a lower dimensional subspace, with an arbitrary size. We then employ a signal recovery algorithm OMP, to obtain parameter updates in the original space. On the other hand, GaLore requires using SVD to obtain the projection matrix for dimensionality reduction, and uses the transpose to recover the original dimensions. In addition, GaLore needs to update the projection matrix every $T$ iterations, whereas SGC does not require this. An important distinction is that GaLore uses SVD, and the compression rate is dependent on the gradient dimensions. However, SGC projects the gradients to an arbitrary small size, independent of the parameter dimensions.
> >
> > > Additionally, MESGC appears to be time-consuming, particularly regarding the practical adjustments outlined in Algorithm 4, and the numerical results in Table 2 do not consistently demonstrate CESGC’s superiority over GaLore.
> >
> > The extra time complexity of MESGC is the result of having to process optimizer states in a lower dimensional subspace with the size independent of the gradient dimension, and have made efforts to address this. Specifically, we implemented a GPU batch efficient implementation of OMP that reduces the runtime by over 10 times compared to a standard implementation that does not utilize the GPU. In Table 2, we show that CESGC can achieve similar performance to other approaches like GaLore while using a smaller number of optimizer states. We also further highlight the benefits of our approach in fine-tuning limited datasets (Section 5.3), in which CESGC outperforms both LoRA and GaLore.
> >
> >   ## Questions
> > > What is the result of the runtime comparison between MESGC and existing PEFT methods?
> >
> > We performed a runtime comparison when fine-tuning LLaMA2-7B on the commonsense 170k dataset for one epoch, on a single H100 GPU. For distributed environments, OMP computation can be further optimized, thereby reducing the gap even more and we leave this as future work.
> >
> > | Method | Runtime (hours) |
> > |-----------|-----------|
> > | Full Fine-tuning | 4.5 |
> > | LoRA | 4.0 |
> > | GaLore | 5.0 |
> > | CESGC | 7.5 |
> > | MESGC | 20.0 |
> >
> >
> > > What is the performance of MESGC on small datasets, as discussed in Section 5.3?
> >
> > We evaluated MESGC on a range of dataset splits of BoolQ as we did for CESGC in our paper. The results show a similar pattern in which our approach outperforms both LoRA and GaLore. The hyperparameters for MESGC are num. chunks=256, sparsity=2, $\kappa$=2, and $\alpha$=2 making it a fair comparison in terms of equal number of optimizer states with LoRA and GaLore.
> >
> > | Dataset Size | Accuracy (LoRa) | Accuracy (GaLore) | Accuracy (MESGC)
> > |-----------|-----------|-----------|-----------|
> > | 500 | 79.1 | 79.4 | 79.4
> > | 1000 | 78.9 | 79.8 | 79.9
> > | 1500 | 78.9 | 79.8 | 80.3
> > | 2000 | 78.8 | 79.6 | 80.7

---

### Note · Authors · 2024-12-16

I have read and agree with the venue's withdrawal policy on behalf of myself and my co-authors.